# *C. elegans* germ granules sculpt both germline and somatic RNAome

Ian F. Price[1,2,3,4], Jillian A. Wagner[1,2,4], Benjamin Pastore[1,2,3], Hannah L. Hertz[1,2] & Wen Tang ®[1,2] ✉

Germ granules are membrane-less organelles essential for small RNA biogenesis and germline development. Among the conserved properties of germ granules is their association with the nuclear membrane. Recent studies demonstrated that LOTUS domain proteins, EGGD-1 and EGGD-2 (also known as MIP-1 and MIP-2 respectively), promote the formation of perinuclear germ granules in *C. elegans*. This finding presents a unique opportunity to evaluate the significance of perinuclear localization of germ granules. Here we show that loss of *eggd-1* causes the coalescence of germ granules and formation of abnormal cytoplasmic aggregates. Impairment of perinuclear granules affects certain germline classes of small RNAs including Piwi-interacting RNAs. Transcriptome profiling reveals overexpression of spermatogenic and cuticle-related genes in *eggd-1* hermaphrodites. We further demonstrate that disruption of germ granules activates HLH-30-mediated transcriptional program in somatic tissues. Collectively, our findings underscore the essential role of EGGD-1 in germ granule organization and reveal an unexpected germ granule-to-soma communication.

Germ cells play a vital role in transmitting genetic and epigenetic information to future generations. Development and maintenance of metazoan germ cells requires electron-dense and membrane-less germ granules[1–6]. Germ granules are described as a hub for RNA metabolism and post-transcriptional gene regulation. Specific components, such as small RNAs, are produced during gametogenesis, deposited into germ granules, and passed on to the offspring. Transmission of germ granules influences gene regulation and phenotypic traits in the progeny. In some cases, the regulatory effect persists for many generations, demonstrated in studies of RNA-induced epigenetic silencing in worms[7–10]. Unlike other membrane-less organelles, germ granules across metazoa preferentially reside at the nuclear periphery. For example, germ granules in *C. elegans* associate with the cytoplasmic face of nuclear pores[11,12]. The association of germ granules with nuclear membranes has also been reported in zebrafish and mammals[13,14]. However, the significance of their perinuclear localization remains not fully understood.

In *C. elegans*, germ granules exhibit a distinctive organization into perinuclear sub-compartments. So far at least four types of sub-compartments have been discovered: P granules, Z granules, SIMR foci, and Mutator foci[11,15–17]. Although these sub-compartments are in close proximity to one another, genetic analyses and proteomic studies revealed distinct components within each sub-compartment[3,6]. Specifically, P granules contain RNA binding protein PGL-1, Piwi proteins (known as PRG-1 in *C. elegans*) and piRNAs (Piwi-interacting RNAs)[18–20]. Z granules, on the other hand, are characterized by the presence of Argonaute protein WAGO-4 (Worm Argonaute protein) and the RNA helicase ZNFX-1[9,16,21,22]. SIMR foci exhibit the enrichment of the tudor domain protein SIMR-1[17]. Mutator foci harbor several mutator proteins, including MUT-7 and MUT-16, which play essential roles in the generation of endogenous siRNAs (small interfering RNAs)[15,23].

Our lab recently employed a proximity-based labeling method in combination with mass spectrometry to define P granule

[1]Department of Biological Chemistry and Pharmacology, The Ohio State University, Columbus, OH 43210, USA. [2]Center for RNA Biology, The Ohio State University, Columbus, OH 43210, USA. [3]Ohio State Biochemistry Program, The Ohio State University, Columbus, OH 43210, USA. [4]These authors contributed equally: Ian F. Price, Jillian A. Wagner. ✉e-mail: tang.542@osu.edu

constituents[24,25]. Among newly identified components, we have characterized several key regulators including LOTUS domain and IDR (Intrinsically Disordered Regions)-containing proteins EGGD-1 and −2 (Embryonic and Germline P Granule Detached), also referred to as MIP-1 and −2 (MEG-3 interacting protein) by Gunsalus's lab[24,26]. EGGD-1 and EGGD-2 act partially redundantly to promote the perinuclear localization of P granules and promote fertility[24,26]. We showed that EGGD-1 is intrinsically capable of self-assembling into perinuclear granules[24]. Remarkably, EGGD-1 is not only necessary but also sufficient to recruit GLH-1, a DEAD-box helicase Vasa homolog, to the nuclear periphery[24,26]. GLH-1 is subsequently required for the proper localization of the scaffold protein PGL-1 to P granules[20,27].

The discovery of LOTUS domain proteins provides a unique opportunity to investigate the importance of the perinuclear localization of germ granules. In this study, we thoroughly characterized phenotypes of *eggd-1* mutants. Specifically, we show that loss of *eggd-1* leads to coalescence of germ granules and mis-localization of P granules, Z granules, and Mutator foci, collectively referred to as PZM granules, within the adult gonad. We find that EGGD-1 is dispensable for RNAi (RNA interference), but essential for the PRG-1/piRNA mediated silencing and production of specific types of endogenous siRNAs. Our small RNA and transcriptome profiling revealed that male-specific small RNA pathways—ALG-3/4 and spermatogenic genes are aberrantly expressed in *eggd-1* hermaphrodites. Unexpectedly, depletion of *eggd-1* or other P granule components causes the nuclear accumulation of the conserved transcription factor HLH-30 in the soma[28,29]. Our results show that activation of HLH-30 enhances the transcription of one cuticle-related gene in somatic tissues. Taken together, our work reveals essential roles of EGGD-1 in small RNA biogenesis and gene regulation and highlights the communication between germ granules and soma.

## Results

### Loss of *eggd-1* causes coalescence of germ granules

*C. elegans* germ granules are known to associate with the nuclear membrane of germ cell nuclei and demix to distinct perinuclear sub-compartments, referred to as PZM granules[3]. Each sub-granule contains specific proteins. For example, PGL-1, ZNFX-1, MUT-16 are enriched at P-, Z-, M- granules, respectively[15,16,21]. While EGGD-1 and EGGD-2 act partially redundantly, EGGD-1 plays a more dominant role in promoting the perinuclear localization of P granules[24,26]. Therefore, we focused on EGGD-1 and interrogated its role in the organization of PZM granules. To this end, we examined the expression of endogenous ZNFX-1, PGL-1, and MUT-16 with fluorescent tags in wild-type and *eggd-1* mutants containing deletion of its full open reading frame[24]. Germ cell nuclei in *C. elegans* are situated along the surface of the gonadal tube and share a common cytoplasmic core called the rachis[30]. Given that the dynamics of PGL-1 foci can change when tagged with different fluorescent proteins[31], we inspected both PGL-1::RFP and PGL-1::GFP fluorescence on the surface and at the rachis of adult gonad. In wild-type animals, PGL-1::RFP or PGL-1::GFP foci were primarily associated with the periphery of germ cell nuclei (Fig. 1a, b and Supplementary Fig. 1a). However, when *eggd-1* was deleted in either *pgl-1::gfp* or *pgl-1::rfp* strains, it resulted in the dispersal of perinuclear PGL-1 and accumulation of PGL-1 aggregates at the rachis (Fig. 1a, b)[24]. The remaining PGL-1::GFP foci at the *eggd-1* nuclear periphery appeared to associate with the nuclear pore protein NPP-11 (Supplementary Fig. 1a)[32]. Of note, the dispersal of PGL-3, another member of the PGL family, was previously observed when *mip-1/eggd-1* was depleted[26]. Furthermore, we sought to examine GFP::ZNFX-1 and GFP::MUT-16 expression and found weaker fluorescence signals at the periphery of germ cell nuclei in *eggd-1* mutants relative to wild-type animals (Fig. 1a, b). Similar to PGL-1::GFP and PGL-1::RFP foci, both GFP::ZNFX-1 and GFP::MUT-16 foci were accumulated at the rachis upon deletion of *eggd-1* (Fig. 1a, b).

We next measured the volume of PZM granules at wild-type nuclear membranes, *eggd-1* nuclear membranes, and *eggd-1* rachis. The mean volume of PGL-1::RFP granules at the *eggd-1* nuclear membrane was 2.64-fold smaller than that of wild-type animals, decreasing from 0.482 $\mu m^3$ to 0.183 $\mu m^3$ (Supplementary Fig. 1b). In contrast, there was a 1.96-fold increase in the mean volume of PGL-1::RFP granules at the *eggd-1* rachis (0.947 $\mu m^3$) compared to that at the wild-type nuclear membrane (0.482 $\mu m^3$) (Supplementary Fig. 1b). This increase was due to the presence of abnormally large PGL-1::RFP granules which could reach up to 25 $\mu m^3$ in the *eggd-1* rachis (Fig. 1b and Supplementary Fig. 1b). We also noted a similar expression pattern for PGL-1::GFP foci. The mean volume of PGL-1::GFP granules at the *eggd-1* nuclear periphery was 2.77-fold smaller than that of wild-type animals, reducing from 0.332 $\mu m^3$ to 0.120 $\mu m^3$ (Supplementary Fig. 1c). While large PGL-1::GFP foci were observed at the *eggd-1* rachis, their size was smaller than cytoplasmic PGL-1::RFP foci (Supplementary Fig. 1b, c). For GFP::ZNFX-1 granules, the mean volume at the *eggd-1* nuclear periphery and rachis decreased 2.32 fold and 1.64 fold respectively, when compared to wild-type perinuclear granules (Supplementary Fig. 1d). On the other hand, the mean volume of GFP::MUT-16 in *eggd-1* mutants only modestly changed compared to wild-type (Supplementary Fig. 1e). We next examined the organization of PGL-1::RFP and GFP::ZNFX-1 granules. Consistent with previous findings[16], GFP::ZNFX-1 granules were separated from and adjacent to PGL-1::RFP foci in wild-type animals (Fig. 1c). Despite reduced signals, PGL-1::RFP and GFP::ZNFX-1 foci remained compartmentalized at the *eggd-1* nuclear periphery (Fig. 1c). At the *eggd-1* rachis, PGL-1::RFP and GFP::ZNFX-1 foci remained separate. Notably, multiple GFP::ZNFX-1 foci were often found to localize to the surface of a single large PGL-1::RFP granule (Fig. 1c). Considering the impact of fluorescent tags on protein phase-transition properties[31], we proceeded to evaluate the organization of PGL-1::GFP and RFP::ZNFX-1 granules. PGL-1::GFP and RFP::ZNFX-1 remained separate at the nuclear peripheries of wild-type and *eggd-1*, as well as at the *eggd-1* rachis (Supplementary Fig. 1f). These findings suggest that P granules and Z granules demix into distinct sub-compartments, regardless of their association with the nuclear membrane.

We next asked how large cytoplasmic granules are formed in the *eggd-1* mutant germ line. To answer this question, we utilized time-lapse DIC and confocal fluorescence microscopy to examine dynamics of PGL-1::RFP. We observed that PGL-1::RFP granules at the *eggd-1* rachis (indicated by yellow and cyan arrows) were more mobile than those at the nuclear periphery (indicated by magenta arrow) (Fig. 1d). Over time, PGL-1::RFP granules at the rachis transited the gonad—originating in mitotic region of the germ line and accumulating in maturing oocytes—presumably through cytoplasmic streaming (Fig. 1d)[33]. Time-lapse microscopy further revealed that large PGL-1::RFP granules were formed by collision and subsequent coalescence of smaller granules in the mitotic regions of the *eggd-1* germ line (Fig. 1e). However, we also noted examples where PGL-1::RFP granules remained in contact but did not coalesce, suggesting that collision between granules does not always result in coalescence (Supplementary Fig. 1g). Taken together, our data suggest that EGGD-1 promotes the perinuclear localization of PZM granules, and that the coalescence of PGL-1::RFP foci can drive the formation of large cytoplasmic granules in the *eggd-1* germ line.

### Loss of *eggd-1* affects a subset of endogenous RNAi pathways

Germ granules host different types of small RNAs[3,5,34]. RdRPs (RNA-dependent RNA Polymerases), such as RRF-1 and EGO-1, produce antisense siRNAs that target many germline expressing transcripts[10,35–37]. These siRNAs are mainly 22-nts (nucleotides) in length and start with 5′ guanine residues, are therefore referred to as 22G-RNAs[34,38]. 22G-RNAs are loaded into distinct AGO (Argonaute) proteins, including CSR-1 and WAGO-1 which localize to P granules, WAGO-4 which localizes to Z granules, the nuclear WAGO-9/HRDE-

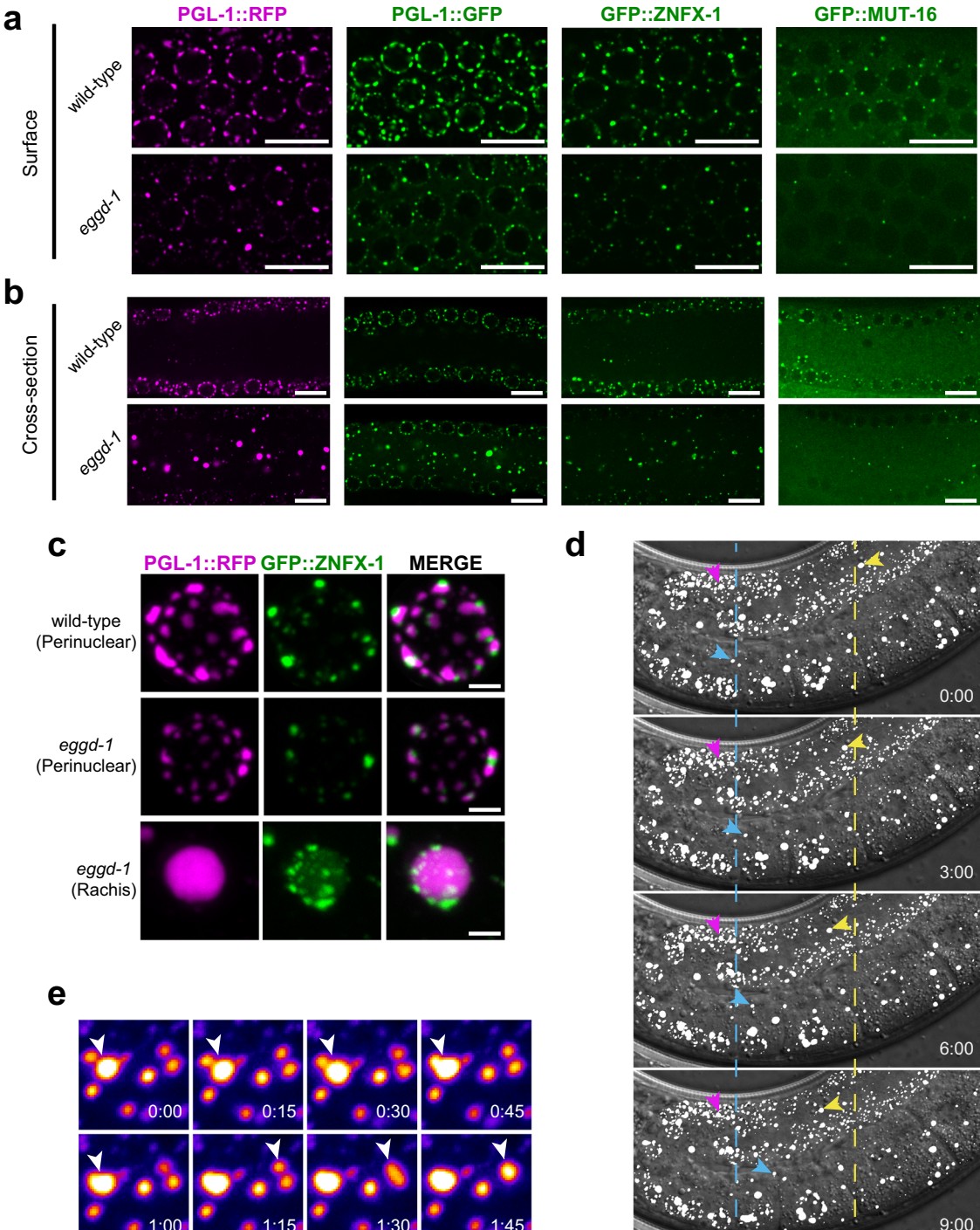

**Fig. 1 | Loss of *eggd-1* leads to coalescence and reorganization of germ granules.** **a** Single confocal slices of fluorescence micrographs visualizing PGL-1::RFP, PGL-1::GFP, GFP::ZNFX-1, or GFP::MUT-16 at the surface of wild-type and *eggd-1* adult germ lines. *n* = 6 independently imaged worms over two independent experiments for each tagged strain. Scale bars = 10 μm. **b** Same as in (**a**). but showing an optical cross-section of the germ line. **c** Maximum intensity projections of confocal fluorescent micrographs showing PGL-1::RFP (magenta) and GFP::ZNFX-1 (green) in wild-type or *eggd-1* mutant germ lines depicting single pachytene nuclei (top two) or a large PGL-1::RFP granule (bottom) localizing to the *eggd-1* rachis. *n* = 6 independently imaged worms over two independent experiments. Scale bar = 2 μm. **d** Time-lapse fluorescence micrographs of a binary maximum-intensity projection PGL-1::RFP (white) confocal z-stack merged with a DIC transmitted light image

showing cytoplasmic streaming of PGL-1::RFP granules in the *eggd-1* germ line. The yellow arrow tracks the streaming of a granule in the pachytene region of the germ line, while the cyan arrow tracks the streaming of a granule in the oocytes. The dashed yellow and cyan lines are references for the starting point of granules tracked by arrows of the respective color. The magenta arrow indicates a perinuclear granule in the late pachytene region that remain relatively stationary. *n* = 2 independent experiments. Time is in minutes:seconds. Scale bar = 20 μm. **e** Time-lapse fluorescence micrographs showing two events of collision and coalescence of PGL-1::RFP foci in the *eggd-1* rachis. Coalescing granules are indicated by white arrows. *n* = 2 independent experiments. Time is in minutes:seconds. Scale bar = 1 μm.

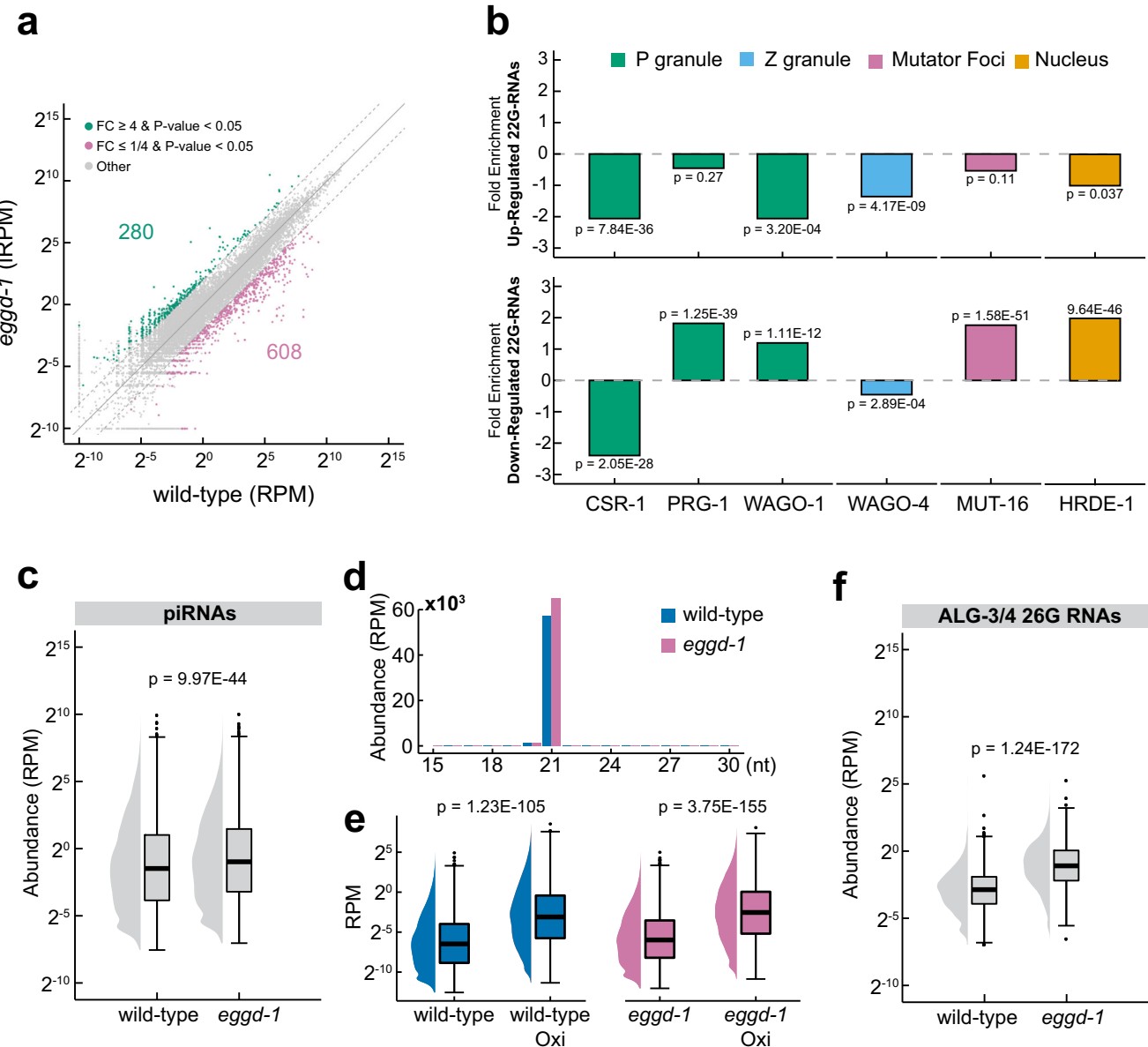

**Fig. 2 | *eggd-1* affects ALG-3/26G-RNA, WAGO/22G-RNA, and PRG-1/piRNA pathways. a** Scatterplot showing the levels of 22G-RNAs mapping to genes in wild-type and *eggd-1* mutants. Genes with four-fold enriched 22G-RNAs

$(\log_2\left(\frac{eggd-1}{wild\ type}\right) \geq 2, p-value < 0.05)$ and four-fold depleted 22G-RNAs

$(\log_2\left(\frac{eggd-1}{wild\ type}\right) \leq -2, p-value < 0.05)$ are shown in green and magenta, respectively. Two-tailed T-test was used to derive *p* values. Top or bottom dotted lines indicate 4-fold change. FC Fold change. RPM Reads per Million. Source data are provided as a Source Data file. **b** Overlap of upregulated and downregulated 22G-RNAs with HRDE-1, CSR-1, PRG-1, WAGO-1, WAGO-4 and MUT-16 targets. The fold enrichment of up/down genes was calculated using the equation:

$\log_2\left(\frac{\%\ of\ genes\ belonging\ to\ the\ i^{th}\ set}{\%\ of\ genes\ belonging\ to\ the\ i^{th}\ set\ among\ all\ genes}\right)$. Fisher's Exact Test was used to derive *p*

values. **c** Boxplot showing the normalized RPM of piRNAs (*n* = 12,859) in wild-type and *eggd-1* mutants. The median piRNA abundance is shown as a solid black line.

Each box displays the interquartile range of the data (between the 25th and 75th percentile). Two-tailed *t* test was used to derive *p* values. Source data are provided as a Source Data file. **d** Length distribution analysis of piRNAs in wild-type (blue) and *eggd-1* mutants (magenta). Shown on the *y* axis is the abundance of piRNA at each length shown on the *x* axis. **e** Boxplots showing the abundance of piRNAs (*n* = 12,859) in wild-type (blue) and *eggd-1* (magenta) under untreated and oxidizing conditions. The median piRNA abundance is shown as a solid black line. Each box displays the interquartile range of the data (between the 25th and 75th percentile). Two-tailed *t* test was used to derive *p* values. Source data are provided as a Source Data file. **f** Boxplots showing the abundance of 26G-RNAs mapping to ALG-3/4 targets in wild-type and *eggd-1* mutants (*n* = 1335, ALG-3/4 target transcripts)[57] is shown as a solid black line. Each box displays the interquartile range of the data (between the 25th and 75th percentile). Two-tailed *t* test was used to derive *p* values. Source data are provided as a Source Data file.

1[22,38–40]. PRG-1/piRNAs recognize hundreds of germline transcripts and recruit RdRPs to initiate production of 22G-RNAs that are loaded into WAGOs[7,8,41–43]. The accumulation of WAGO class 22G-RNAs depends on MUT-16, a key component of Mutator foci[15,44].

We sought to determine whether aberrant PZM granules in *eggd-1* mutants lead to the mis-regulation of small RNAs. To this end, we cloned and sequenced small RNAs from synchronized wild-type and

*eggd-1* mutant adults. Our analysis revealed that 608 transcripts exhibited downregulated 22G-RNAs and 280 transcripts exhibited upregulated 22G-RNAs in *eggd-1* mutants relative to wild-type (4-fold-change, *p* value < 0.05, Two-tailed t-test test) (Fig. 2a). To determine the affected AGO pathways in *eggd-1* mutants, we categorized differentially expressed 22G-RNA loci based on the AGOs they associate with or the factors their biogenesis depends on (Fig. 2b). CSR-1 22G-RNAs

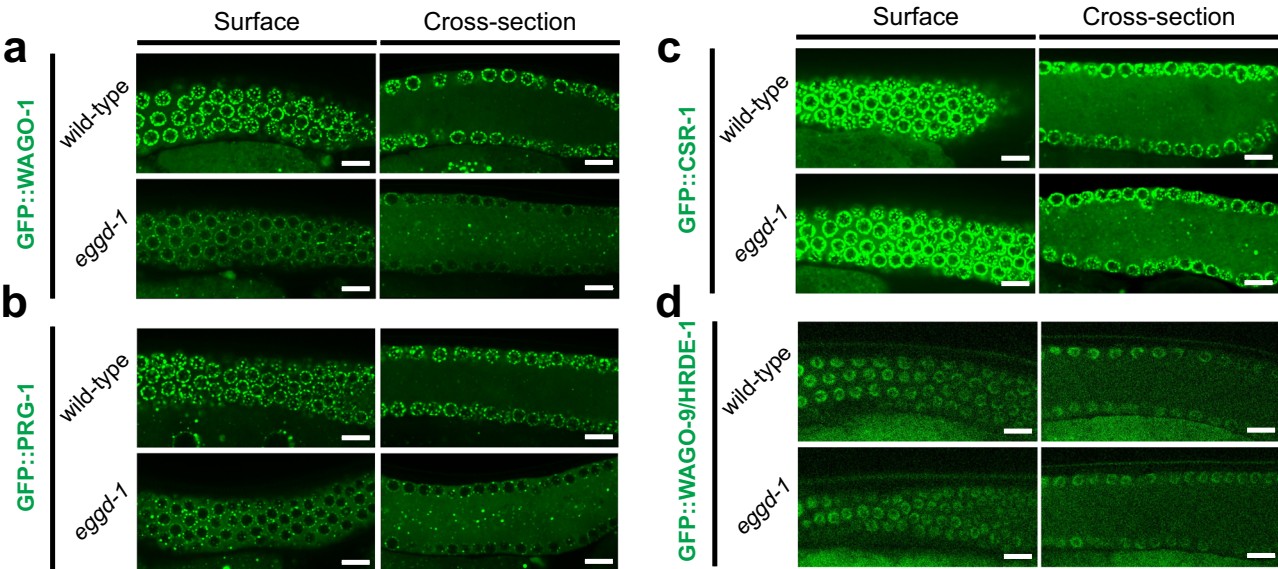

**Fig. 3 | Mis-localization of Argonaute proteins in *eggd-1* mutants.** Single plane confocal images showing the expression of GFP::WAGO-1 (**a**), GFP::PRG-1 (**b**), GFP::CSR-1 (**c**), and GFP::WAGO-9/HRDE-1 (**d**) at the surface (left) or cross-section (right) of the wild-type and *eggd-1* gonad. *n* = 5 independently imaged worms over one experiment. Scale bar = 10 µm.

and WAGO-4 (Z granule specific AGO) class 22G-RNAs were underrepresented in both upregulated and down-regulated gene sets, suggesting neither small RNA pathway was strongly disrupted in *eggd-1* animals. In contrast, WAGO-1 class 22G-RNAs (1.4-fold enrichment), HRDE-1/WAGO-9 class 22G-RNAs (2.0-fold enrichment), PRG-1 dependent 22G-RNAs (1.9-fold enrichment), and MUT-16 dependent 22G-RNAs (1.8-fold enrichment) were overrepresented in the down-regulated set, but underrepresented in the upregulated set (Fig. 2b). These data indicate that EGGD-1 is required for the production of 22G-RNAs in the WAGO pathway.

*C. elegans* piRNAs are derived from over 15,000 discrete piRNA producing loci[19,45–47]. After 5′ and 3′ processing, mature piRNAs have a uniform length of 21-nts, start with a 5′ monophosphorylated uridine, and are 2′ O-methylated at the 3′ end[48–51]. We found that the overall abundance of piRNAs was slightly increased in *eggd-1* mutants, with a 1.56-fold increase in median abundance compared to wild-type (Fig. 2c). In *C. elegans*, specific piRNAs are enriched in male or female germ lines[52,53]. While the abundance of female and gender-neutral piRNAs was comparable between *eggd-1* mutants and wild-type, male-specific piRNAs were elevated in *eggd-1* mutants. (Supplementary Fig. 2). Because PARN-1 (the piRNA 3′ trimming enzyme) and HENN-1 (The piRNA 3′ methyltransferase) are enriched in perinuclear granules[48,54], we next examined if piRNA 3′ processing is altered upon loss of *eggd-1*. In both wild-type and *eggd-1* animals, piRNAs are 21-nts in length (Fig. 2d). To determine the methylation status of piRNAs, we carried out sodium periodate-mediated oxidation experiments and sequenced oxidation-resistant small RNAs. The vicinal diol at 3′ termini of unmodified RNAs can be oxidized to a dialdehyde, rendering them poor substrates for small RNA cloning whereas RNAs with terminal 2′-O-methylation are resistant to oxidation. piRNAs in both wild-type and *eggd-1* animals are comparably enriched by oxidation when normalized to total number of reads (Fig. 2e). Taken together, our results indicate that piRNA biogenesis is mostly unaltered in *eggd-1* mutants, but there is an abnormal accumulation of male-specific piRNAs.

The unexpected finding that *eggd-1* hermaphrodites aberrantly express male piRNAs prompted us to examine male-specific ALG-3/4 siRNA pathways. Argonaute proteins—ALG-3 and ALG-4—are loaded with 26G-RNAs that target a set of male transcripts[55–57]. ALG-3/4 class 26G-RNAs are exclusively expressed during spermatogenesis and are

required for male fertility at elevated temperatures[55–57]. In wild-type animals, the median abundance of 26G-RNAs per gene was 0.13 RPM (Reads per Million) (Fig. 2f). However, their median abundance was increased to 0.47 RPM when *eggd-1* was mutated (Fig. 2f). We conclude that male specific piRNAs and ALG-3/4 26G-RNAs are also expressed in *eggd-1* hermaphrodites.

## Expression of Argonaute proteins in *eggd-1* mutants

Next, we investigated whether the expression of individual AGOs is affected by loss of *eggd-1*. To this end, We introduced a full *eggd-1* open reading frame deletion allele into strains of *prg-1*, *wago-1*, *wago-9/hrde-1*, and *csr-1*, which were endogenously tagged with GFP[58]. Germ granules have reduced retention at nuclear periphery and become dispersed at the rachis when *eggd-1* was depleted (Fig. 1)[24,26]. Therefore, to assess the impact of *eggd-1* loss on each GFP::AGO protein, we quantified GFP signals of individual germ nuclei and signals at the rachis (Supplementary Fig. 3a, b). Given that AGO proteins exhibit varying expression levels, we adjusted the exposure times and laser intensities accordingly to capture the individual GFP::AGO proteins, while maintaining the identical experimental conditions between the wild-type and *eggd-1* strains. We found less perinuclear GFP::WAGO-1, GFP::PRG-1 and GFP::CSR-1 in *eggd-1* mutants compared to wild-type (Fig. 3a, b, c and Supplementary Fig. 3a). Specifically, the median intensity of GFP::WAGO-1, GFP::PRG-1 and GFP::CSR-1 reduced by 1.34-, 1.38- and 1.22-fold, respectively (Supplementary Fig. 3a). In contrast, the nuclear GFP::WAGO-9/HRDE-1 signals were comparable between wild-type and *eggd-1* mutants (Fig. 3d and Supplementary Fig. 3b), even though GFP::WAGO-9/HRDE-1 class 22G-RNA levels were reduced (Fig. 2b). When quantifying the rachis versus total signals from cross-section images of the gonad, we observed a general increase from strains expressing GFP::WAGO-1, GFP::PRG-1, GFP::CSR-1 upon loss of *eggd-1* (Fig. 3a, b, c and Supplementary Fig. 3b). Notably, not all the AGOs are affected to the same degree in *eggd-1* mutants. For example, GFP::PRG-1 exhibited the strongest accumulation in the rachis (median ratio was increased from 0.336 in wild-type to 0.570 in *eggd-1* animals) (Fig. 3b and Supplementary Fig. 3b), while GFP::CSR-1 showed only a modest accumulation in the rachis (Fig. 3c and Supplementary Fig. 3b). Taken together, our findings suggest that EGGD-1 is required for proper perinuclear localization of AGOs, particularly the PRG-1 protein.

**a**

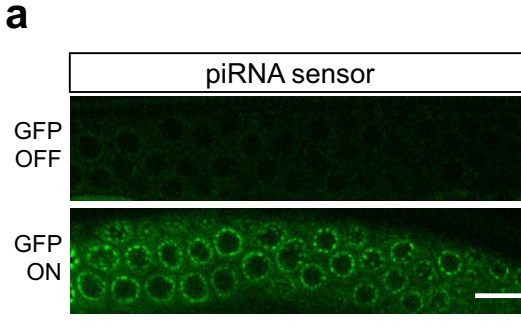

**b**

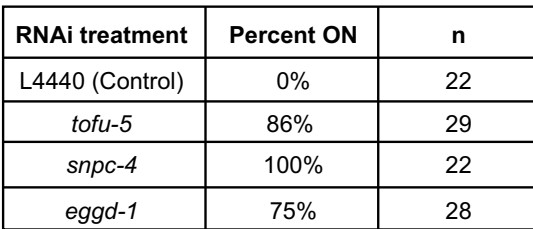

| RNAi treatment | Percent ON | n |
|---|---|---|
| L4440 (Control) | 0% | 22 |
| *tofu-5* | 86% | 29 |
| *snpc-4* | 100% | 22 |
| *eggd-1* | 75% | 28 |

**c**

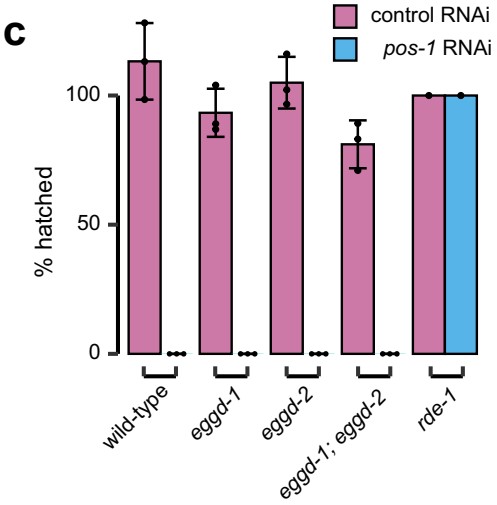

**d**

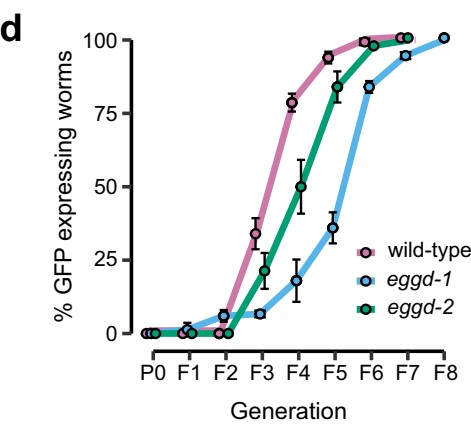

**Fig. 4 | EGGD-1 is required for piRNA-mediated gene silencing, but dispensable for gene silencing by dsRNAs. a** Representative single plane confocal images of the piRNA reporter strain when exposed to control (top) and *snpc-4* (bottom) RNAi food. *n* for each RNAi condition was shown in Fig. 4b. Scale bar = 10 μm. **b** Table summarizing the percentage of worms with reporter ON for each RNAi treatment. *n* = worms imaged over one experiment. **c** Bar graph showing mean ± standard deviation percentage of embryos hatched for wild-type, *eggd-1, eggd-2,* and *eggd-1; eggd-2* worms plated on L4440 (control) or *pos-1* RNAi (*n* = 3 independent experiments). *rde-1* dataset was collected from one experiment. For each experiment, *n* ≥ 200 embryos assayed for viability. Source data are provided as a Source Data file. **d** Line graph showing GFP RNAi inheritance in three independent experiments for *eggd-1, eggd-2,* and wild-type worms over eight successive generations. Error bars indicate mean ± standard deviation. 50 worms per line were imaged each generation to determine GFP silencing status. Source data are provided as a Source Data file.

## EGGD-1 is required for piRNA-mediated gene silencing

The finding that PRG-1 is mis-localized in *eggd-1* mutants prompted us to investigate if *eggd-1* mutants are defective in piRNA-mediated transgene silencing. Through base-pairing interactions, *C. elegans* piRNAs target newly introduced transgenes and some endogenous transcripts, and initiate 22G-RNA production[7,8,41,42,59]. PRG-1/piRNAs complexes are required for initiating the silencing of transgenes, but not for maintaining it[7,8,41,42]. We employed an established piRNA sensor strain to determine the role of *eggd-1* in initiation of piRNA-mediated silencing[60,61]. The piRNA sensor was 100% silenced in wild-type germ line (Fig. 4a)[60,61]. However, when transcription factors involved in piRNA expression, such as TOFU-5 and SNPC-4, were depleted by RNAi[62–64], the sensor became de-silenced (Fig. 4a, b). Depletion of *eggd-1* by RNAi also led to de-silencing of the piRNA sensor (Fig. 4b), indicating that EGGD-1 is necessary for piRNA-mediated silencing.

To further test if EGGD-1 is involved in the initiation or maintenance of piRNA-mediated silencing, we used reporter strains with *gfp::h2b* transgenes containing or lacking a piRNA target site[8]. Consistent with previous studies[8], *gfp::h2b* became silenced only when the piRNA target site was present (Supplementary Fig. 4a, b). We introduced *prg-1* or *eggd-1* mutations to the silenced transgene by genetic crosses and propagated the strains for approximately 16 generations. In the absence of *prg-1*, the reporter remained 100% silenced. Similarly, the reporter was not de-silenced in *eggd-1* mutants (Supplementary

Fig. 4b, c). Therefore, once transgene reporter became silenced, neither *prg-1* nor *eggd-1* is required for maintaining silencing. Combined with the observations of reduced PRG-1-dependent 22G-RNAs (Fig. 2b) and mis-localization of PRG-1 (Fig. 3b) in *eggd-1* mutants, our findings suggest that perinuclear PRG-1/piRNAs may be crucial for the initiation of piRNA-mediated silencing.

## EGGD-1 is dispensable for initiation and inheritance of RNAi

After identifying defects in the expression of germline AGOs (Fig. 3), we next asked if *eggd-1* mutants are defective in RNAi (RNA interference)[65]. In *C. elegans*, RNAi can be induced experimentally by feeding worms bacteria that express dsRNAs (double stranded RNAs) targeting specific transcripts[66]. To test RNAi competency, we exposed worms to dsRNA against the *pos-1* gene, which must be expressed maternally to support embryonic development[67]. Exposure of wild-type animals to *pos-1* dsRNA led to 100% embryonic lethality (Fig. 4c)[68]. Consistent with previous findings[68], deletion of *rde-1*, which encodes the primary AGO required for exogenous RNAi, rendered animals fully resistant to *pos-1* RNAi (Fig. 4c). Mild embryonic lethality was observed in *eggd-1* and *eggd-1; eggd-2* mutant animals exposed to control RNAi (Fig. 4c)[26]. Upon *pos-1* dsRNA treatment, *eggd-1, eggd-2,* and *eggd-1; eggd-2* double mutants produced no viable progeny (Fig. 4c). We conclude that *eggd-1* and *eggd-2* mutants are RNAi competent and that germ granule integrity may not be essential for RNAi.

In *C. elegans*, the introduction of dsRNAs can trigger gene silencing that can persist for multiple generations, even in the absence of further dsRNA exposure, a process known as RNAi inheritance[40,69,70]. To evaluate the role of EGGD proteins in dsRNA-induced RNAi inheritance, we employed a reporter strain that expresses a *gfp::h2b* transgene under a germline specific promoter[10,16,71,72]. We crossed *eggd-1* and *eggd-2* mutants to the reporter strain. The *eggd-1; eggd-2* double mutants were excluded from this analysis due to their severe germline atrophy and rapid decline in fertility over generations[24,26]. As expected, exposure of wild-type animals to *gfp* dsRNAs led to the complete silencing of *gfp::h2b* at the P$_0$ generation. After dsRNA triggers were removed, *gfp::h2b* were gradually de-silenced. The silencing lasted for up to ~5 generations (Fig. 4d)[10,16,71,72]. Similar to wild-type animals, *eggd-1* and *eggd-2* mutants displayed fully penetrant GFP::H2B silencing at the P$_0$ generation. Surprisingly, GFP::H2B silencing persisted longer in *eggd*-1 and *eggd-2* mutants, suggesting enhanced transgenerational RNAi inheritance (Fig. 4d). Such prolonged RNAi heritance was also observed in *prg-1* mutants[73]. Taken together, our findings indicate that the LOTUS domain proteins EGGD-1 and EGGD-2 limit RNAi inheritance.

## Spermatogenic genes are upregulated in *eggd-1* hermaphrodites

The finding of the aberrant expression of small RNAs in *eggd-1* mutants motivated us to profile their transcriptome. To this end, we prepared and sequenced ribosomal RNA depleted RNA samples from synchronized wild-type and *eggd-1* mutant adults. Applying DESeq2 for differential expression analysis[74], we identified 1024 upregulated transcripts and 133 downregulated transcripts (including the *eggd-1* transcript itself) in *eggd-1* animals compared to wild-type (fourfold-change, *p* adjusted value < 0.05) (Fig. 5a). Two analyses were conducted to investigate the upregulated gene dataset (*n* = 1024). First, we performed tissue enrichment analysis[75], which revealed a significant enrichment of genes involved in male development (Fig. 5b). Second, we compared our dataset with a previously published study in which spermatogenic, oogenic, and gender-neutral germ line-expressed genes were defined[76]. Notably, there was a considerable overlap of *eggd-1* upregulated genes with spermatogenic genes, while there were few overlaps with oogenic genes (Fig. 5c). It was intriguing that certain spermatogenic genes were significantly upregulated while some appeared unchanged in *eggd-1* mutants (Fig. 5a, c). To further investigate this finding, we plotted the fold-change of transcripts in *eggd-1* relative to wild-type against the fold-change of transcripts in male (*fem-3*) compared to female (*fog-2*) (Supplementary Fig. 5a). This analysis revealed a strong positive correlation (Pearson correlation, *r* = 0.68, *p* value < 0.05). Transcripts highly enriched in male (*fem-3*) displayed significant upregulation in *eggd-1* compared to wild-type, while transcripts with modest enrichment in male (*fem-3*) showed modest or no change (Supplementary Fig. 5a). These global analyses revealed a general upregulation of spermatogenic genes in *eggd-1* mutants, which we further validated by examining individual genes. For instance, *alg-3*, which encodes a male-specific AGO, and *gsp-3*, which encodes a sperm-specific PP1 phosphatase are exclusively expressed in male (*fem-3*) (Fig. 5d, e and Supplementary Fig. 5a). Both genes showed strong upregulation in *eggd-1* mutants (Fig. 5d, e and Supplementary Fig. 5a)[55,56,77]. In contrast, *spe-44*, a gene encoding a transcription factor[78], is highly expressed in male (*fem-3*) but also modestly expressed in female (*fog-2*). We found that the expression of *spe-44* was unaffected upon loss of *eggd-1* (Supplementary Fig. 5a, b).

In light of the upregulation of spermatogenic genes, we conducted experiments to determine whether *eggd-1* mutants exhibited the MOG phenotype (masculinization of the germ line). We cultured synchronized wild-type and *eggd-1* L1 larvae and subsequently examined the germ line of L4/adult worms using brightfield microscopy as well as fluorescence microscopy with DAPI staining to aid in the visualization of sperm. As a positive control, we employed RNAi against *mog-4*, a gene encoding a DEAH-Box protein[79]. We used the following criteria to determine the MOG phenotype: 1. Completion of vulval development. 2. Absence of oocytes, and 3. Presence of excess sperm in the germ line. While the majority of *mog-4* dsRNA treated animals displayed the MOG phenotype, none of the wild-type animals exhibited it (Fig. 5f, g). Out of the 570 *eggd-1* worms that were imaged, we only identified one hermaphrodite exhibiting the MOG phenotype and one male (Fig. 5f, g). These findings suggest that the upregulation of spermatogenic genes in *eggd-1* mutants is unlikely to be a result of the masculinization of the germ line.

## Cuticle genes are upregulated in *eggd-1* hermaphrodites

Our tissue enrichment analysis showed that genes expressed in the epithelial system were upregulated in *eggd-1* mutants (Fig. 5b). This finding was corroborated by Gene Ontology (GO) enrichment analysis[80]. In our examination of the upregulated genes in *eggd-1* mutants (*n* = 1024), we found that the most enriched GO term in the domain of Biological Process was "*structural constituent of cuticle*" (Fig. 6a). We constructed the interaction network of *eggd-1* upregulated genes using publicly available protein-protein interaction data[81]. The resulting network contained 987 nodes and 69,888 non-redundant edges (Supplementary Fig. 5c). The number of edges was significantly higher than expected edges by chance (*p* value < 1.0$^{-16}$), suggesting that these genes are biologically connected as functional groups. Indeed, we identified two main clusters: one enriched for genes involved in spermatogenesis and the other enriched for cuticle-related genes (Supplementary Fig. 5c). Taken together our analyses reveal that germ granule integrity represses the expression of spermatogenic genes and cuticle-related genes.

## HLH-30 is involved in germline-to-soma communication

The observation of overexpressed cuticle-related genes upon *eggd-1* mutation is intriguing, given that EGGD-1 is primarily expressed in the gonad and germ lineage of embryos[24,26]. The question arose as to whether the aberrant overexpression of cuticle-related genes was occurring in the germ line or in somatic tissues. To answer this question, we monitored the expression of collagens—major components of *C. elegans* cuticles. We used an established *col-12p::DsRed* transcriptional reporter in which *DsRed* is expressed under the *col-12* promoter[82]. Wild-type young adults displayed weak DsRed signals, whereas *eggd-1* mutants exhibited significantly higher signals in both the hypodermis and intestine (Fig. 6b). Consistent with this finding, the transcript level of *col-12* was significantly increased in *eggd-1* mutants relative to wild-type.

To determine the molecular mechanism for *col-12* upregulation, we sought to identify transcription factor(s) associated with the *col-12* promoter and potentially drive its transcription. To this end, we examined chromatin immunoprecipitation-sequencing (CHIP-seq) datasets of a wide variety of transcription factors from the modENCODE project[83]. This unbiased search revealed that transcription factors DAF-16 and HLH-30 occupy the *col-12* promoter under the physiological condition (Supplementary Fig. 6a). By analyzing published HLH-30 CHIP-seq data[83], we defined 5158 potential targets of HLH-30. The promoters of 29 genes that are associated with cuticle development exhibited binding of HLH-30 (Supplementary Fig. 6b). Additionally, in our analysis of transcription factor binding site enrichment[80], we noted a significant overrepresentation of the HLH-30 binding motif in the dataset of genes upregulated in *eggd-1* mutants (*n* = 1024) (Supplementary Fig. 6c). Among the upregulated gene set, the promoters of 94 genes displayed HLH-30 occupancy (Supplementary Fig. 6b). These findings prompted us to assess the expression and activity of HLH-30 and DAF-16. Transcription factors HLH-30 and DAF-16 are widely expressed in many tissues and diffused within the cell. In response to external stresses or germline ablation, both

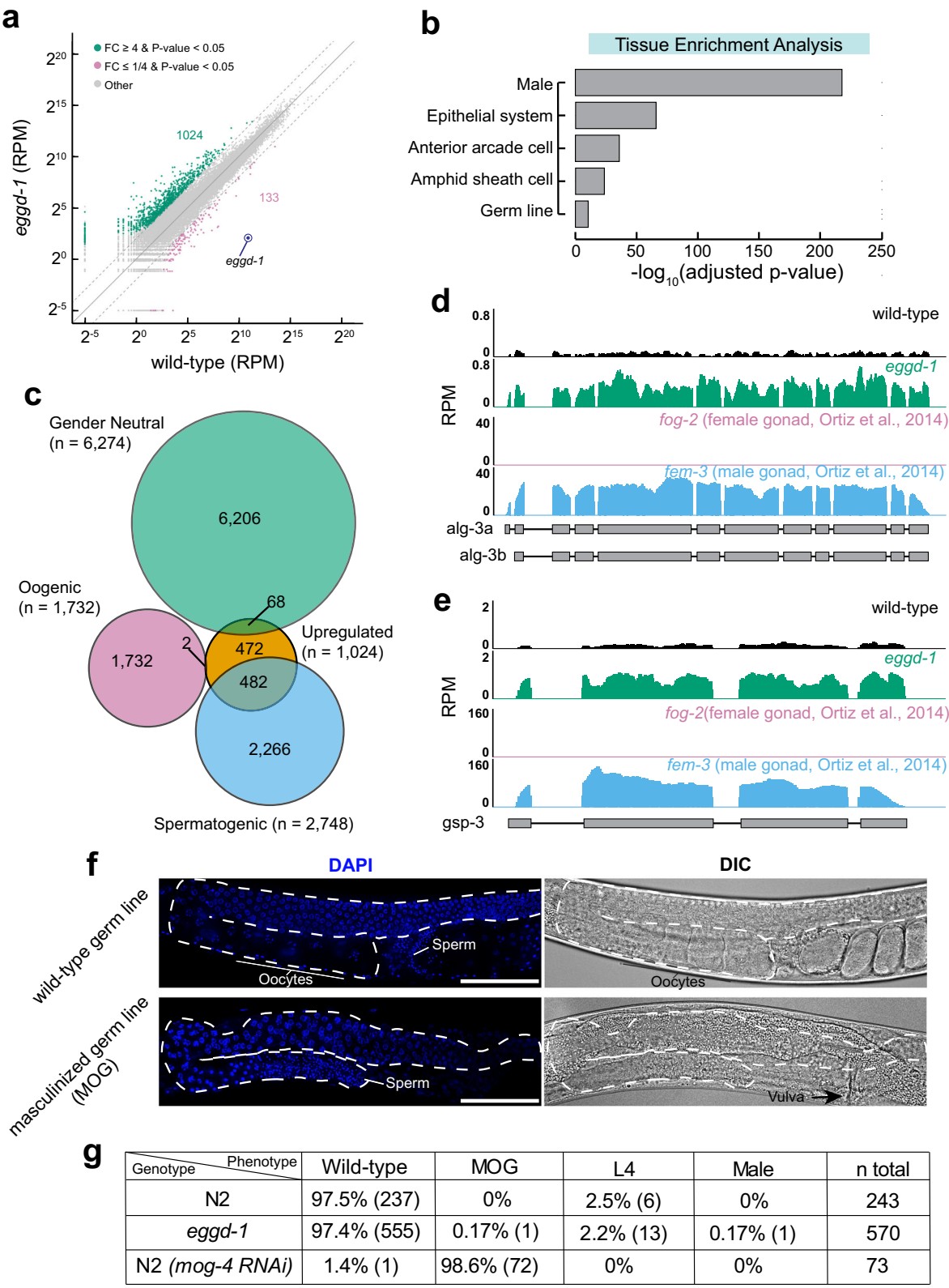

transcription factors enter the nuclei of somatic cells including intestinal cells and neurons[28,29]. By promoting the transcription of their target genes, HLH-30 and DAF-16 play essential role in stress responses and lifespan extension[28,29,84]. We obtained CRISPR/Cas9-engineered strains expressing DAF-16::mKate2 or HLH-30::GFP[85,86], and examined their expression in the intestine—an easy-to-score and functionally relevant tissue[84]. Consistent with previous reports[28,29], both DAF-16::mKate2 and HLH-30::GFP were present at a low level in intestinal

nuclei of wild-type young adults (Fig. 6c and Supplementary Fig. 6d). DAF-16::mKate2 translocated to the nucleus under heat stress, but did not accumulate upon depletion of *eggd-1* (Supplementary Fig. 6d). In contrast, deletion or depletion of *eggd-1* led to elevated levels of HLH-30::GFP in intestinal nuclei, but not in germ nuclei (Fig. 6c). We next wished to determine if the accumulation of nuclear HLH-30 is responsible for activation of *col-12* in *eggd-1* mutants. We used RNAi to deplete *hlh-30* in the presence of the *col-12p::DsRed* reporter and

**Fig. 5 | Spermatogenic and cuticle-related genes are upregulated in *eggd-1*
hermaphrodites. a** Scatter plot showing the level of transcripts in wild-type and
*eggd-1* as measured by RNA sequencing. Genes with fourfold upregulation
($\log_2\left(\frac{eggd-1}{wild\ type}\right) \geq 2$, $p-value < 0.05$) are shown in green. Genes with four-fold
downregulation ($\log_2\left(\frac{eggd-1}{wild\ type}\right) \leq -2$, $p-value < 0.05$) are shown in magenta. Two-
tailed T-test was used to derive *p* values. Top or bottom gray dotted lines indicate
fourfold change. *eggd-1* transcript itself is shown as a blue dot and labeled. FC Fold
change. RPM Reads per Million. Source data are provided as a Source Data file.
**b** Tissue enrichment analysis of genes significantly upregulated 4-fold in *eggd-1*
mutants compared to that in wild-type. Benjamini-Hochberg adjusted *p* values was
indicated at *x* axis. **c** Overlap of significantly upregulated genes in *eggd-1* mutants
compared to that in wild-type with spermatogenic (*n* = 2748), oogenic (*n* = 1732),
and gender neutral genes (*n* = 6274) defined by RNA sequencing of isolated female

(*fog-2*) and male (*fem-3*) gonad[76]. **d** Browser track showing the abundance of *alg-3*
transcript as measured by RNA sequencing from wild-type and *eggd-1* animals as
well as *fog-2* and *fem-3* gonad[76]. The coverage shown is the average of two
sequencing replicates (wild-type and *eggd-1*) or seven sequencing replicates (both
*fog-2* and *fem-3* mutants). **e** Same as in (**d**), however showing the *gsp-3* locus.
**f** Examples of wild-type and MOG adult phenotypes used to classify worms under
the conditions listed in (**g**) imaged by DIC transmitted light or DAPI fluorescence to
visualize DNA. The germ line is outlined by a dashed line in each image. *n* for
representative wild-type and MOG worms are listed in (**g**). **g** Table showing the
percentage and count (*n*) of relevant phenotypes exhibited by N2, *eggd-1*, and *mog-
4* RNAi-treated worms over two (N2 and *eggd-1*) or one (*mog-4*) independent
experiments.

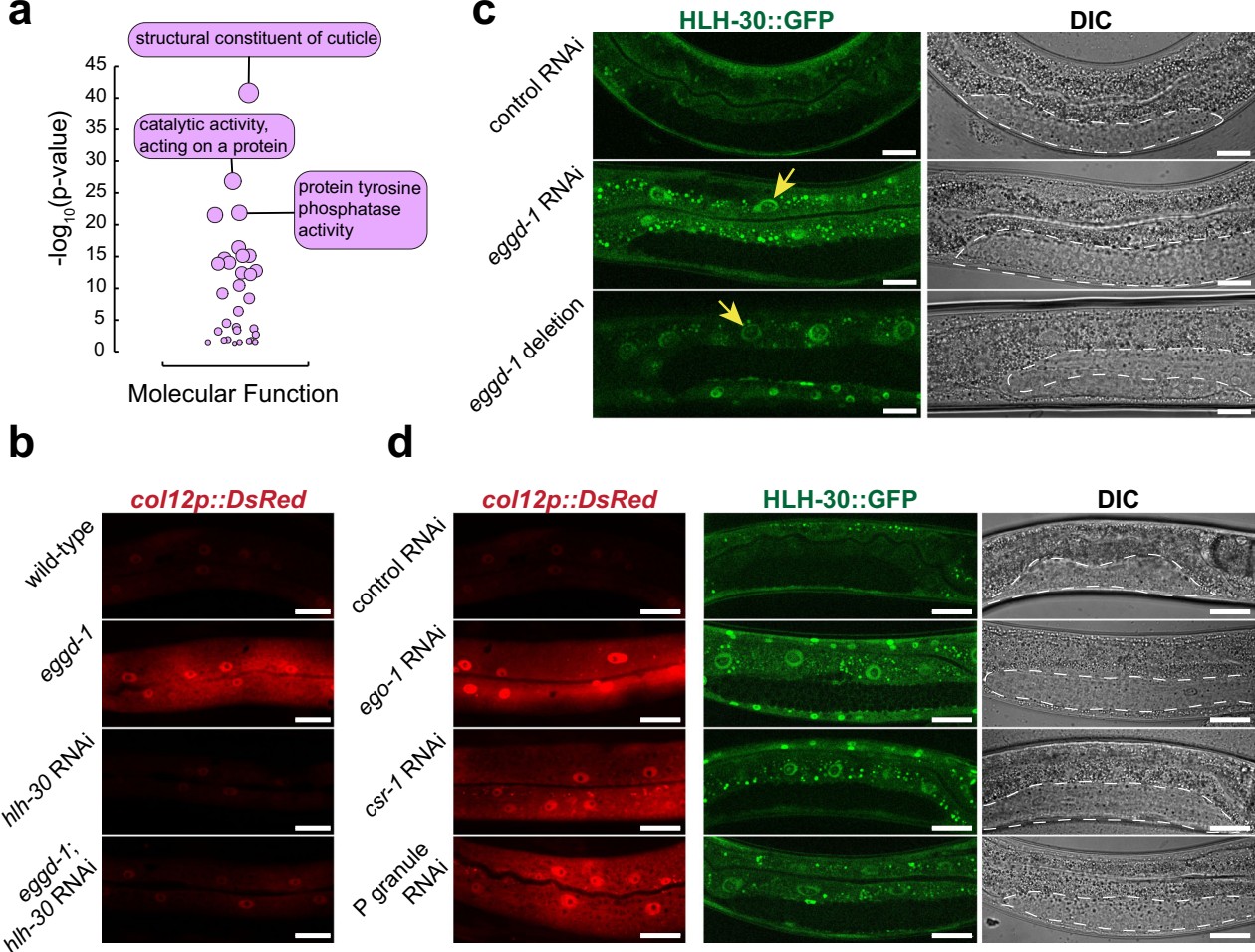

**Fig. 6 | HLH-30 participates in communication between germ granules and
soma. a** Gene ontology analysis of transcripts significantly upregulated in *eggd-1*
mutants. Shown on the *y*-axis is the $-\log_{10}$ (*p* value) of the significance of the
overlap with upregulated genes and gene ontology classes. The top three GO
categories for molecular function (purple) are labeled. **b** Maximum intensity pro-
jections of col-12p::DsRed expression in the hypodermis of wild-type and *eggd-1*
adults, plated on either OP50 or *hlh-30* RNAi *n* = 8 independently imaged worms
over two experiments. Scale = 10 µm. **c** Single plane images of intestinal HLH-
30::GFP expression with control (L4440) RNAi and *eggd-1* RNAi, and in *eggd-1* null

mutants. White dashed lines outline the gonad the DIC images (right). GFP images
(left) depict HLH-30::GFP intestinal signal. Arrows indicate intestinal nuclei. Scale
bar = 20 µm. *n* = 8 independently imaged worms over two independent experi-
ments. **d** Maximum intensity projections of hypodermal col-12::DsRed reporter
expression in worms exposed to control (L4440), *ego-1*, *csr-1*, or P granule RNAi
(left). Single plane DIC (right) images and accompany fluorescence images of HLH-
30::GFP (middle) upon individual RNAi treatments. P granule RNAi: *pgl-1, pgl-3, glh-
1, glh-4* RNAi. Scale = 20 µm. *n* = 8 independently imaged worms over two inde-
pendent experiments.

found that knockdown of *hlh-30* lowered DsRed levels in *eggd-1*
mutant animals (Fig. 6b).

Next, we asked whether defective P granules in general activates
HLH-30 and/or elicit upregulation of collagen. Previous work showed
that depletion of *csr-1* or *ego-1* led to the accumulation of large and

irregular P granules[39,87,88]. Furthermore, P granule can be efficiently
depleted by simultaneously targeting four main P granule components
*glh-1, glh-4, pgl-1,* and *pgl-3*, a strategy referred to as P granule RNAi[89].
Upon exposure to *csr-1* RNAi, *ego-1* RNAi, or P granule RNAi, the *col-
12p::DsRed* reporter displayed an increase in *DsRed* fluorescent signals

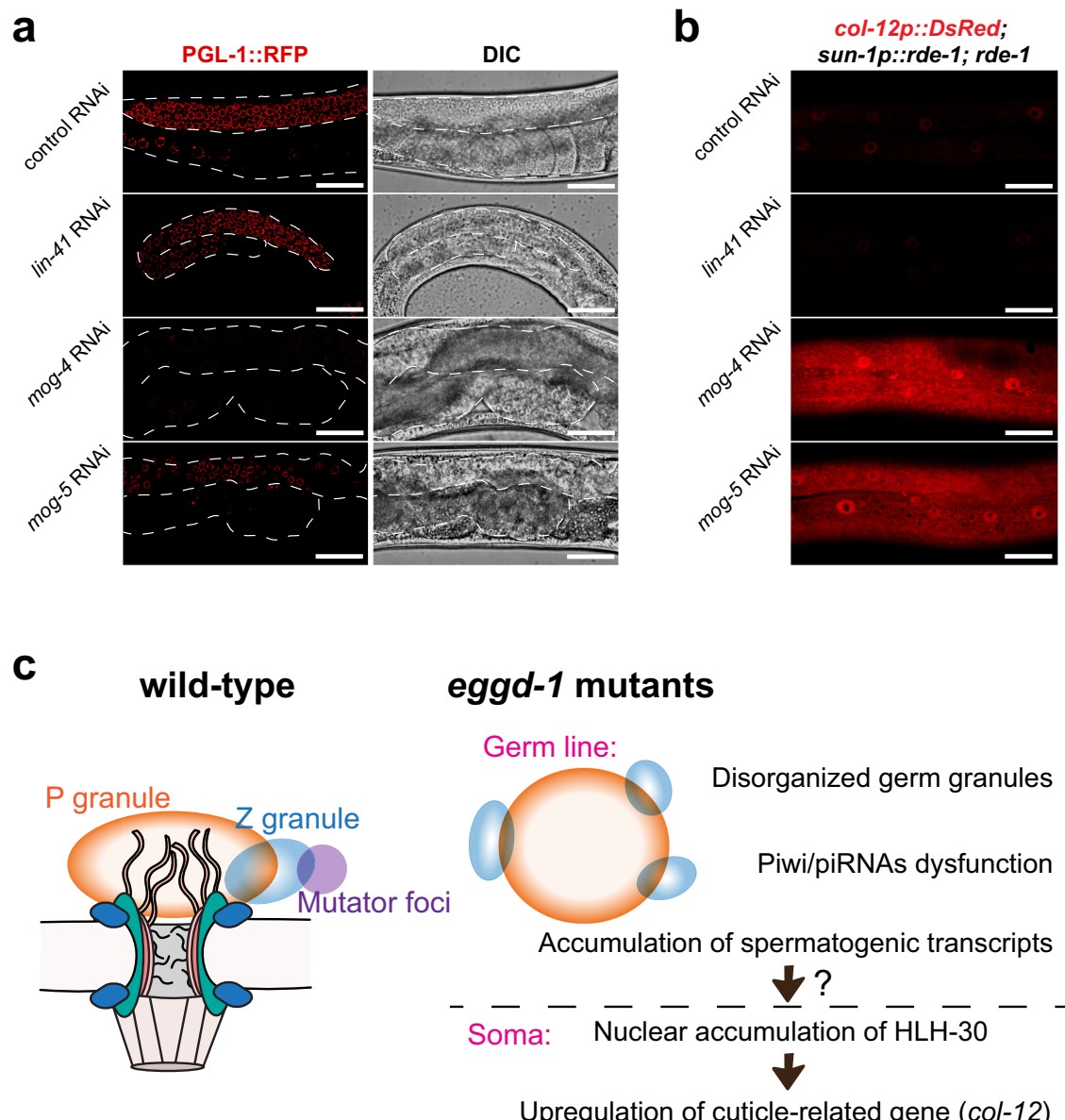

**Fig. 7 | Aberrant expression of spermatogenic gene may enhance somatic COL-12 expression. a** Single plane DIC and confocal images of *pgl-1::rfp* worms exposed to control (L4440), *lin-41*, *mog-4*, or *mog-5* dsRNAs. White dashed lines outline the gonad. Scale = 20 μm. *n* = 10 independently imaged worms over one experiment. **b** Maximum intensity projections of hypodermal col-12p::DsRed reporter expression in germline RNAi competent worms exposed to control (L4440), *lin-41*, *mog-4*, or *mog-5* dsRNAs. Scale = 20 μm. *n* = 12 independently imaged worms over two independent experiments. **c** Model illustrating the role of EGGD-1 in the organization of germ granules, and gene regulation in both germ line and somatic tissues.

when compared to control RNAi. Furthermore, All three RNAi treatments led to accumulation of nuclear HLH-30::GFP in the intestine, but not in the germ line (Fig. 6d). Collectively, our findings suggest that proper formation of germ granules acts to suppress the accumulation of nuclear HLH-30 and repress the expression of collagen genes such as *col-12* in the soma.

We next sought to determine the underlying cause of HLH-30 activation and somatic *col-12* overexpression in *eggd-1* mutants. Previous and current studies have demonstrated the crucial role of EGGD-1 in germline maintenance, oocyte development, and suppression of spermatogenic genes in hermaphrodites (Fig. 5)[24,26]. This led us to speculate whether defects in oogenesis and/or activation of spermatogenic genes could be responsible for the abnormal *col-12* expression. To explore this idea, we generated the *sun-1p::rde-1; col-12p::DsRed; rde-1* strain in which the endogenous *rde-1* gene is mutated and a

single-copy *rde-1* transgene is expressed under the germline-specific *sun-1* promoter[90]. This strain, which is somatic RNAi deficient but germline RNAi competent, allowed us to examine the effects of germline-specific knockdown[90]. Consistent with a previous report[91], depletion of LIN-41, a TRIM-NHL protein, resulted in a decrease in the size of germ line and complete loss of oocytes (Fig. 7a). However, P granule formation, indicated by PGL-1::RFP expression, appeared unaffected by *lin-41* depletion (Fig. 7a). Depletion of *lin-41* in *sun-1p::rde-1; col-12p::DsRed; rde-1* strain did not alter *col-12p::DsRed* expression (Fig. 7b), indicating defective oogenesis is not responsible for *the col-12* activation. We next induced the expression of male genes in hermaphrodites by depleting key regulatory proteins in the sex determination pathway, including DEAH-box proteins MOG-4 and MOG-5[79]. A previous study has shown that loss of *mog-4*, or *mog-5* leads to the germline masculinization[79]. Indeed, we observed disorganized

and masculinized germ line upon the depletion of *mog-4* or *mog-5* (Fig. 7a). Furthermore, *mog-4* knockdown significantly reduced PGL-1::RFP levels, while *mog-5* knockdown modestly reduced PGL-1::RFP expression (Fig. 7a). Upon exposure to *mog-4* or *mog-5* dsRNAs, the *sun-1p::rde-1; col-12p::DsRed*; *rde-1* reporter displayed increased *DsRed* fluorescent signals (Fig. 7b). Taken together, our findings suggest that aberrant expression of spermatogenic genes in hermaphrodites may contribute to the elevated somatic expression of *col-12*.

## Discussion

Germ granules across species are characterized as perinuclear structures[1–6]. High-resolution microscopy revealed that germ granules in *C. elegans* are separated into distinct sub-compartments[15–17]. The unique perinuclear organization of germ granules raises several questions. Why and how are germ granules associated with nuclear membrane? How are sub-compartments formed and maintained? LOTUS domain containing proteins have emerged as pivotal regulators in germ granule organization. *Oskar*, containing both LOTUS and OSK domains, organize germ plasms in *Drosophila*[92]. TDRD5 and TDRD7 possess LOTUS and Tudor domains and are indispensable for germ granule formation in mice[93–95]. Recent studies conducted on *C. elegans* established an essential role of LOTUS domain proteins in facilitating perinuclear localization of germ granules[24,26,96]. Here, using a series of microscopy and genetic experiments, we demonstrated that the LOTUS domain protein EGGD-1 is required for the proper formation of germ granule sub-compartments including P granules, Mutator foci and Z granules (Fig. 7c). Upon loss of *eggd-1*, large PGL-1 aggregates are accumulated at the rachis of gonad, possibly resulting from the coalescence of small cytoplasmic granules. Although these large PGL-1 aggregates are in close contact with one or multiple ZNFX-1 foci, PGL-1, and ZNFX-1 granules remain largely separated (Fig. 1). Our findings suggest that the association of germ granules with the nuclear membrane, possibly through nuclear pores, prevents their coalescence. Moreover, demixing of germ granules into sub-compartments is independent of their association with the nuclear membrane.

Our study, which combines small RNA profiling and genetic experiments, suggests that the perinuclear localization of germ granules promotes small RNA biogenesis and transcriptome surveillance. A specific set of AGO proteins and key RNA processing enzymes are enriched at *C. elegans* germ granules[3,4]. It has been proposed that these AGO proteins and their small RNA co-factors, including piRNAs and 22G-RNAs, surveil transcripts as they exit the nucleus. Such surveillance recognizes and silences non-self sequences and permits the expression of germline expressing genes[7,8,41–43]. Consistent with this idea, our findings indicate that PRG-1/piRNA mediated surveillance was compromised in *eggd-1* mutants, presumably because PRG-1/piRNA failed to robustly localize to perinuclear germ granules (Fig. 7c). On the contrary, exogenous RNAi appears to be intact, or even enhanced over generations in *eggd-1* mutant animals (Fig. 4c). Three silencing pathways engaging nuclear WAGO protein HRDE-1, perinuclear WAGO-1 and WAGO-4, and cytoplasmic WAGO proteins including RDE-1 and PPW-1, are thought to promote robust exogenous RNAi responses[9,22,34,38,97]. Thanks to these partially redundant silencing pathways, it is possible that the integrity of perinuclear granules is dispensable for RNAi. Furthermore, we envision that depletion of endogenous 22G-RNAs upon *eggd-1* deletion can free up WAGOs and postulate that surplus WAGOs can be loaded with exogenous 22G-RNAs to elicit enhanced RNAi responses.

*C. elegans* hermaphrodites produce both sperm and oocytes. The germ line initiates spermatogenesis at the L4 larval stage, and then switches to oogenesis during the adult stage. Previous studies reported that EGGD-1 is essential for germline development and fertility[24,26]. Here our transcriptome and small RNA profiling data revealed that male specific small RNAs and a set of spermatogenic genes are upregulated in *eggd-1* mutant adults. While it is plausible that loss of *eggd-1*

leads to incomplete shutdown of spermatogenesis during the sperm-to-oocyte switch, we cannot rule out the possibility of reactivation of the spermatogenesis program during oogenesis. Recent studies suggest that the expression of spermatogenic mRNAs during sperm-to-oocyte switch is repressed by Piwi protein PRG-1, AGO protein CSR-1, WAGO-pathway factor RDE-3, Mutator foci factor MUT-16 and additional P granule components[44,60,98,99]. These findings, combined with our work, provide a unifying model that germ granules and endogenous siRNA pathways play a crucial role in promoting the sperm-to-oocyte transition. Failure to suppress spermatogenesis during oogenesis perhaps leads to defective germ cells, which contributes to infertility.

Our study uncovered communication between the germ line and soma that results from germ granule stress (Fig. 7c). It has been proposed that *C. elegans* germ granules play a crucial role in repressing the expression of somatic genes in the germ line and prevent germ cells from undergoing somatic development[89,100]. This model describes germ granules as the safeguard for germline transcriptome. To our surprise, we found that depletion of key germ granule components such as *eggd-1*, *csr-1* or *ego-1* causes the accumulation of HLH-30 in the nucleus of somatic cells, but not in the germ nuclei (Fig. 6). Using a transcriptional reporter monitoring collagen expression[82], we showed that the nuclear accumulation of HLH-30 promotes the expression of at least one collagen gene in soma. HLH-30 and its mammalian homolog TFEB translocate to the nucleus in response to various stresses including nutrient deprivation, heat stress and germline ablation[28,29]. By regulating autophagy-related and aging-related genes, HLH-30 is essential for lifespan extension and stress responses[28,29]. The lack or disorganization of germ granules in the germ line may constitute some form of stress, which triggers the accumulation of HLH-30 in somatic nuclei. Nuclear HLH-30 activates its targets such as some collagen genes, which may translate distress signals into developmental decisions. In future work, it will be exciting to explore the physiological roles of germ granules in stress responses, stress resistance, and longevity.

## Methods
### Strains
Worms were grown under standard conditions at 20 °C unless otherwise indicated[101]. The Bristol N2 strain serves as wild type. The list of strains used in this study is supplied in the Supplementary Data File 1.

### CRISPR genome editing
Deletion of *eggd-1* CDS in *gfp::wago-9* to make WHY612 was achieved using CRISPR-Cas9 genome editing[24]. Two CRISPR guide RNAs (IDT) targeting the open reading frame of *eggd-1* (upstream: GACATTC ACTTGGCAAATGA; downstream: TCGGATAAGGATAGTTGGTG) were injected with a single-stranded DNA donor: ATGTTCTCTCAG AAGTGACATTCACTTGGCAAATGCTCGAGTGATCGAAATTTTTACGTGC TTTAAAATATCCTGTT. The vector pRF4 containing a dominant allele of *rol-6* was used as a co-injection marker[102]. Roller F1 were picked and the *eggd-1* deletion was screened for by PCR.

### RNAi by dsRNA feeding
The HT115 RNAi feeding strains were picked from the *C. elegans* Ahringer RNAi Collection (Source Biosciences). NGM plates containing 50 μg/ml ampicillin and 5 mM IPTG were seeded with HT115 bacteria expressing dsRNAs against the target gene. Worms synchronized by bleaching were plated on the RNAi plates and imaged after 2–3 days. To assay germline RNAi competence, ~100 synchronized L1 larvae per strain were plated to *pos-1* RNAi or L4440 RNAi plates until adulthood. After ~80 h, 15 adults were transferred to a single NGM plate in three replicates for 3 h and then removed. Laid eggs were counted, and the subsequent day hatched F1 were counted to score embryonic viability. RNAi inheritance was assayed by plating 3 replicates of

200 synchronized L1 reporters to NGM plates seeded with HT115 expressing GFP dsRNA at 20 °C. Approximately 60 h after plating, 50 young adult worms of each genotype were imaged to score silencing of a GFP::H2B transgene in the germ line. The remaining animals were used to obtain a synchronized population by hypochlorite lysis. Subsequent generations were propagated on NGM plates seed with OP50. Each generation, 50 worms per replicate were scored as GFP-off (silenced) or GFP-on (de-silenced) by fluorescence microscopy. Some *eggd-1* and *eggd-2* animals were excluded from analysis due to a complete lack of germ cell nuclei.

## Microscopy

Live animals were immobilized in M9 medium containing 2 mM levamisole and 0.1 μm polystyrene beads (Polysciences) and mounted on glass slides with 5% agar pads. Widefield fluorescent images were acquired using a Nikon Ti2 inverted microscope equipped with a Hamamatsu ORCA-Fusion C14440 detector using a Plan Apo 40×/0.95NA objective in NIS-Elements AR 5.41.02. Spinning disc confocal images were taken in MetaMorph version 7.10.4.452 on a Nikon TiE inverted microscope equipped with an Andor Revolution WD spinning disc system and a pco.Edge bi 4.2 sCMOS detector using a CFI Plan Apo VC 60×/1.2NA water immersion objective or a CFI Plan Apo VC 100×/1.4NA oil immersion objective at room temperature (21–23 °C). For co-localization and granule volume quantification experiments, special care was taken to avoid imaging-associated stress and temperature fluctuations that can affect granule assembly[103,104]: worms were kept in incubators at standard growth conditions until immediately before mounting on slides. Slides were then imaged at an ambient temperature of 19–21 °C for no longer than 10 min.

## Quantification of Argonaute protein expression

GFP::PRG-1, GFP::WAGO-1, GFP::WAGO-9/HRDE-1, and GFP::CSR-1 samples were prepared following the procedures described in the Microscopy section. Z stacks were acquired using the 60x water objective with 1 μm step size. Different exposure times and laser intensities were applied to capture individual GFP::AGOs, while identical parameters were applied between the wild-type and paired *eggd-1* strains. In ImageJ[105], the background signal was determined by selecting an area outside of animals, and subsequently subtracted from the images. To quantify Argonaute protein levels, germ nuclei on the surface of the gonad were selected and the mean intensity of GFP signals were measured. This process was repeated for 25 nuclei per animal. To access the ratio of rachis signals over to total signals, cross sections were extracted from Z stacks. The entire pachytene and rachis regions were selected, and integrated densities were measured to quantify the relative abundance of rachis signals in comparison to the overall signal intensity.

## Imaging and processing of col-12p::DsRed, hlh-30::gfp, and piRNA sensor

The reporters in wild-type and mutant backgrounds, as well as those subjected to with RNAi treatments, were prepared following the procedures outlined in the Microscopy section. Z stacks were acquired using a 60x water objective with a 1 μm step size. Using ImageJ[105], the background signal was determined by measuring the average intensity of an area outside of the worm, and this background signal was subtracted from all z planes. An average intensity projection was used to capture col-12p::DsRed signals. A single plane from the z stack was used to create images of HLH-30::GFP. A maximum intensity projection was created to capture signals of piRNA sensors.

## Quantification of germ granule volumes

Confocal z stacks with a 0.2 μm step size were acquired using a 100× objective with the microscope described in the Microscopy section of the methods. For perinuclear granule volume quantification, 15 nuclei

from 3 separate animals were selected for PGL-1::GFP, PGL-1::RFP, GFP::ZNFX-1, and GFP::MUT-16 wild-type and *eggd-1* mutants. In ImageJ[105], background signal was subtracted by measuring the average intensity of a region of interest outside the worm and subtracting the ROI (Region Of Interest) mean value from the image. Segmentation thresholds were determined using background-subtracted wild-type nuclei by generating a maximum intensity projection of the z stack of the nucleus and using the auto threshold function with the Li threshold algorithm. The 3D simple segmentation plugin was then used to apply the median threshold value of the wild type nuclei to each nuclear z-stack. The resulting segmented images were added to the 3D ROI Manager plugin and the Measure 3D function was used to determine the volume of each granule. To determine the volume of granules localizing to the rachis, a region of interest was drawn to encompass the rachis in four separate germ lines which were segmented and measured as described above. Granule volumes were exported as.csv files and plotted in R using the ggplot2 package.

## Analysis of masculinization of the germ line (MOG) phenotype

N2 and *eggd-1* animal were synchronized by hypochlorite lysis and plated to NGM seeded with OP50 or HT115 expressing *mog-4* dsRNAs (N2 only). N2 and *eggd-1* worms were harvested after 72 h of growth at 20 °C. Due to slow growth, *mog-4* RNAi-treated worms were harvested at 96 h after plating. Harvested worms were suspended with M9 buffer, washed twice with M9 buffer, then twice with ddH₂O and transferred to −20 °C methanol for 20 min. The worms were then transferred to 100% acetone for 20 min at −20 °C, ddH₂O for 20 min at 4 °C, and PBS for 20 min at 4 °C. PBS was removed and Worms were suspended in Vectashield mounting medium with DAPI (Vector Laboratories) and transferred to slides for imaging. Transmitted light and DAPI fluorescence images were acquired using a Nikon Ti2 inverted microscope at 20× magnification. To evaluate for a MOG phenotype, each worm imaged was inspected for 1. Completion of vulval development. 2. Absence of oocytes, and 3. Presence of excess sperm in the germ line. Worms that fulfilled all criteria were scored as MOG.

## Total RNA extraction

Approximately 50,000 synchronized adult worms were collected with M9 using a glass Pasteur pipette. The worms were washed once with M9, rocked in fresh M9 for 5 min, washed once in ddH₂O, and resuspended suspended in 1 ml of TRI Reagent (Sigma). Worms were lysed using the Bead Mill 24 homogenizer (Thermo Fisher Scientific) and the lysate was transferred to a clean tube 1.5 ml centrifuge tube. 0.1 volumes of bromochloropropane were added to the mix and shaken by hand for 30 s. The mixture was then centrifuged at 15,000 × g and the aqueous phase was transferred to a fresh 1.5 ml microcentrifuge tube, followed by the addition of 1 volume of isopropanol. The tube was incubated at −20 °C for 1 h to precipitate RNA and centrifuged for 15 min at 15,000 × g at 4 °C. The supernatant removed by pipetting and the pellet was washed twice by adding 1 mL 75% ethanol. The pellet of RNA was allowed to dry for 3 min to remove remaining ethanol, dissolved in ddH₂O and stored at −80 °C.

## Small RNA cloning

Small RNAs were enriched using MirVana miRNA Isolation Kit (Thermo Fisher Scientific), according to the manufacturer's instructions. β−elimination was performed as described[54]. Small RNA-sequencing libraries were constructed following an previously established protocol[106]. Briefly, Small RNA samples were treated with the polyphosphatase PIR-1 to remove γ and β phosphates from 5′-triphosphorylated RNAs[107]. The monophosphorylated RNAs were ligated to a 3′ adapter (5′rAppAGATCGGAAGAGCACACGTCTGAACTCCAGTCA/3ddC/3′, IDT) using T4 RNA ligase 2 (NEB) at 15 °C overnight. T4 RNA ligase 1 (NEB) was used to ligate the 5′ adapter (rArCrArCrUrCrUrUrUrCrCrCrUrArCrArCrGrArCrGrCrUrCrUrUrCrCrGrArUrCrU, IDT). The

ligated products were converted to cDNA using SuperScript IV Reverse Transcriptase (Thermo Fisher Scientific). The cDNAs were further amplified by PCR, and small RNA libraries with unique barcodes were pooled and sequenced using the Novaseq platform (SP 2 × 50 bp, Illumina).

## mRNA library preparation

Ribosomal RNAs were depleted using the RNase H method[108]. 3 µg total RNA were incubated with antisense DNA oligonucleotides targeting *C. elegans* and *E. coli* ribosomal RNAs, then treated with Hybridase Thermostable RNase H (Biosearch Technologies) at 45 °C for 30 min. The rRNA-depleted samples were treated with Turbo DNase (Thermo Fisher Scientific) at 37 °C for 30 min. RNAs longer than 200 nts were enriched using RNA Clean & Concentrator-5 (Zymo Research). mRNA sequencing libraries were constructed using Ultra II Directional RNA Library Prep Kit (NEB), according to the manufacturer's instructions. The library samples with unique barcodes were pooled and sequenced at the Illumina HiSeq 4000 platform (PE 150 bp).

## Bioinformatic analysis of small RNA sequencing data

Raw small RNA sequencing reads were trimmed of adapters and low-quality sequences using TrimGalore[109] and then collapsed with a custom shell script. Collapsed reads were aligned to a reference containing structural RNA annotations including snRNA, snoRNA, tRNA, and ncRNA using the default Bowtie parameters. Reads that failed to align to structural RNAs were aligned to the *C. elegans* reference genome (WormBase Release WS279) using Bowtie with the parameters −v 0 −m 1000 −a−best −strata[110]. Aligned reads were then normalized to the number of locations mapped, and assigned to genomic features using BEDTools intersect and custom python scripts[111]. Sequences mapping to piRNA loci were required to map uniquely and perfectly (0 mismatch) in the sense orientation, contain a 5′ T, be 15-40 nucleotides in length, have a 5′ end mapping to the 0th, 1st, or 2nd, position of the piRNA annotation. 22G-RNAs were defined as sequences mapping to protein coding gene exons, pseudogene exons, lincRNAs, rRNAs, or transposable elements in the antisense orientation, with a 5′ G or A and 21−23 nucleotides long. 22G-RNA sequences were also required to map perfectly but were allowed to multi-map to 1000 locations. Reads mapped to rRNAs were given preference over other genomic features in the event that the rRNA locus overlaps with another gene. After counting, reads were normalized to the number of features mapped to normalize for overlapping genes. The read count for each gene was aggregated for each genetic background and normalized based on the depth of sequencing (number of reads mapping to the genome) and scaled to 1,000,000 reads. Differential gene expression between genetic backgrounds was preformed using custom Python scripts, all plots were drawn in R or Python. Bed files of assigned reads were converted to BigWig files using BEDOPS and visualized using IGV[112,113]. In addition to genomic alignments collapsed reads were aligned to RepBase transposon consensus sequences[114]. The pipeline used in the smRNA sequencing analysis is available at https://github.com/benpastore/nextflow_smRNA.

## Bioinformatic analysis of mRNA sequencing data

Raw paired-end fastq files were trimmed of adapters and poorly sequenced reads using TrimGalore[109]. Trimmed fastq files were aligned to the *C. elegans* reference (WormBase Release WS279) with STAR, using the parameters−alignIntronMax 10000−outFilterMismatch NoverReadLmax 0.04−outFilterIntronMotifs RemoveNoncanonical−outFilterMultimapNmax 10[115]. Reads were further filtered using SAMTools to remove orphan reads (reads in which one mate in a pair does not align but the other does)[116]. Transcript abundance was then quantified using FeatureCounts using the parameters -p -s 2 -g gene_id -M --fraction[117]. To visualize coverage along the genome BAM files were

converted to BigWig (BW) files using bamCoverage from the package DeepTools using the parameters−normalize CPM−smoothLength 10−binSize 5−exactScaling−filterRNAstrand forward. To visualize replicates BW files were averaged across replicates using bigWigMerge and bedGraphToBigWig from the UCSC Genome Browser Utilities. To quantify transposon sequence abundance we used Salmon tool[118]. Unnormalized counts from Salmon and FeatureCounts were concatenated using custom python scripts. Normalization and differential gene expression analysis was conducted using DESeq2[74]. Data were visualized using R ggplot2 package and IGV. The comprehensive mRNA sequencing pipeline used to trim, align and quantify mRNAs is available at Github: https://github.com/benpastore/nextflow_pipelines.

## Definition of HLH-30 targets

Published CHIP-seq dataset (SRP028480) were used to define potential target genes of HLH-30. Raw sequencing reads were trimmed from adapters, and poorly sequenced reads were removed using TrimGalore[109]. Processed reads were aligned to the *C. elegans* genome (WormBase release WS279) using BWA[119]. MACS2 was then used to call CHIP-seq peaks using a *q* value cutoff of 0.1[120]. HLH-30 targets were defined by assigning CHIP peaks to genomic features by intersecting peak regions to promoters of protein coding gens, lincRNAs and pseudogenes using Bedtools[111]. Promoters of genes were defined as 1 kb upstream of the 5′ UTR start position. Promoters of genes that fall within operons were defined as 1 kb upstream of the 5′ UTR of the first gene in the operon. CHIP-targets were further filtered to include targets that are enriched in both HLH-30 CHIP-seq replicates. The CHIP-seq pipeline used to trim, align and quantify CHIP sequencing data is available at Github: https://github.com/benpastore/nextflow_pipelines.

## Definition of gene classes

To calculate the representation of up/down regulated siRNAs in *eggd-1* mutants we preformed gene enrichment analysis using six gene sets. These gene sets are as follows: 4012 CSR-1 targets[39], 1419 PRG-1 targets[44], 1350 WAGO-1 targets[121], 4572 WAGO-4 targets[22], 1969 MUT-16 targets[44], and 1273 HRDE-1 targets[40].

## Network analysis of protein-protein interaction

The list of upregulated genes in *eggd-1* mutants was submitted to the STRING web server string-db.org using default settings to construct protein-protein interaction network. The network was then exported to Cytoscape for visualization.

## Statistics and reproducibility

Figure 1 microscopy experiments were performed twice with similar results. Figure 3 confocal microscopy experiments were performed once and agree with results from separate widefield microscopy experiments. The piRNA sensor experiment in Fig. 4a, b was performed once. Results from RNAi experiments in Fig. 4c, d are composites of three experiments. Figure 5f, g is composite of two experiments with similar results. RNAi and microscopy experiments in Fig. 6 were repeated twice with similar results. RNAi and microscopy experiments in Fig. 7a were performed once, and Fig. 7b were performed twice with similar results. mRNA sequencing and small RNA sequencing experiments were performed using two biological replicates which correlate well and cluster together in a principle component analysis. No statistical method was used to predetermine sample size. No data were excluded from the analyses. The experiments were not randomized. The Investigators were not blinded to allocation during experiments and outcome assessment. Tests used to assign statistical significance are indicated in figure legends. All attempts at reproducing experimental results were successful.

## Reporting summary

Further information on research design is available in the Nature Portfolio Reporting Summary linked to this article.

## Data availability

Small RNA sequencing and mRNA sequencing data generated in this study have been deposited to NCBI GEO under accession code GSE228857 and GSE228858, respectively. All other supporting data are present in the 'Source Data' file or available upon request. A table with strains used in this study can be found in Supplementary Data File 1. Source data are provided with this paper.

## Code availability

All Scripts used in data analysis are available at Github: https://github.com/benpastore/eggd_RNA_2023. Alternatively, code used for small RNA sequencing is available at https://doi.org/10.5281/zenodo.8302724; code used in mRNA sequencing analysis is available at https://doi.org/10.5281/zenodo.8302720; code use for all other analyses is available at https://doi.org/10.5281/zenodo.8302718.

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

## Acknowledgements

We thank D. Updike for providing bacteria expressing P granule dsRNAs; C. Mello for providing GFP::WAGO-1, GFP::HRDE-1/WAGO-9 and GFP::CSR-1 strains; G. Seydoux for sharing NPP-11::wrmScarlet strain; D. Schoenberg for providing feedback on manuscript; A. Ozturk for designing antisense oligonucleotides against ribosomal RNAs; the Neuroscience Imaging Core for instruments (S10OD010383); the OSU Comprehensive Cancer Center genomics core for Illumina sequencing, Ohio Supercomputer Center for Computational resources, Cae-norhabditis Genetics Center for providing some of the *C. elegans* strains (P40OD010440). This work was supported by The Center for RNA Biol-ogy Fellowship to I.P. and B.P., and NIH Maximizing Investigators' Research Award (R35GM142580) to W.T.

## Author contributions

I.P., J.W., and W.T. designed and conducted experiments; B.P. analyzed small RNA sequencing and mRNA sequencing data; H.H. assayed for RNAi competency and RNAi inheritance. I.P., J.W., B.P., and H.H. generated the figures and wrote figure legend. I.P. and W.T. wrote the manuscript. W.T. supervised the project and obtained research funding. All authors discussed the results and commented on the manuscript.

## Competing interests

The authors declare no competing interests.
