## [Peer Review File · Nature Communications]

C. elegans germ granules sculpt both germline and somatic RNAomeREVIEWER COMMENTS

Reviewer #1 (Remarks to the Author):

Germ granules are conserved features of early germ cells that have been postulated to promote germline identity and prevent germ cells from adopting somatic fates. In this work, the authors further examine intrinsically disordered LOTUS domain proteins, EGGD-1, and EGGD-2 (also known as MIP-1 and MIP-2). These proteins have been previously shown by the authors and others to function partially redundantly in perinuclear granule formation in the *C. elegans* germline. That work defined a role for the IDR domains in organizing and anchoring granules to the nucleus and the LOTUS domain in localizing P- granules proximal to the nucleus; however, the other granule types present in germ cells were not examined in detail. Because prior work indicated EGGD-1 supports granule formation but is more important for perinuclear P-granule localization, this study focuses primarily on EGGD-1 functions. Specifically, the authors investigated the importance of proper subcellular localization of various germ cell granules, P-granules, Z-granules, and M- (mutator) granules and how localization contributes to germ cell development.

Prior work suggested that EGGD-1 and 2 act as scaffolds to recruit and regulated developmental transitions by regulating RNA processing machinery and interacting with nuclear pore components. Here loss of *eggd-1* is examined more closely and leads to coalescence of germ granules in embryos and abnormal localization of PZM granules in adults, supporting a role for EGGD-1 in promoting association of various germ granules with nuclear pores or nuclear membrane/tethers granules maintaining granule spatial distribution. Further, they examined the effects on RNA regulation machinery and provide evidence that EGGD-1 is not broadly required for piRNA biogenesis/pathways but is required for PRG-1 silencing and select endogenous siRNAs. In *C. elegans* hermaphrodites, the developing gonad produces sperm first and later switches to oocyte production. Here the authors analysis of transcriptomes and small RNAs provides evidence that RNAs involved in spermatogenesis persist in hermaphrodites in the absence of EGGD-1, suggesting proper granule localization facilitates RNA regulation at this developmental transition. Further, they observed prolonged transgenerational inheritance of RNAi, as also occurs in *prg-1* mutants, one of the improperly localized factors in *egg1* mutants.

By knocking down or mutating different genes required for proper P-granule size or formation, the authors discovered a generalized role of P-granules, rather than a specific role of EGGD-1, in preventing nuclear accumulation of the transcription factor HLH-30 in the adjacent somatic cells and activation of cuticle genes in adjacent somatic tissues. This surprising observation suggests improper subcellular localization of granules or molecules regulated in these subcellular compartments impacts communication between germline and somatic cells and influences somatic cell development. Overall, the manuscript is well written, and the data are of high quality and are clearly presented. However, a few points related to the exciting observation regarding granules and communication with the soma should be addressed.

Major:

- 1) A potential role for granules in regulating gene expression in the soma is exciting and unexpected. While investigating the precise mechanism would be beyond the scope of this work, an addressable question that was unclear is whether the soma is detecting a molecule regulated by the perinuclear granules or instead is recognizing that the hermaphrodite germline/oocytes are abnormal or dying. Is the same effect on somatic/cuticle gene expression observed if oocytes are abnormal for reasons unrelated to germ granules or are ablated, or if one ectopically expressed spermatogenesis genes in oocytes?
- 2) The methods corresponding to the reporter analyses, including argonauts, HLH-30, Col-12p are not clear or are missing from the methods section. How were these signals normalized across conditions?

Minor:

In some cases, the numbers of animals/cells examined is unclear because the data are not

consistently presented. In some figures numbers are found on the panels or in the legends and in other figures the numbers are in the methods. It would be helpful to present these numbers more consistently.

The abbreviation PZM is used in the introduction but is only fully introduced later.

Reviewer #2 (Remarks to the Author):

The recent discovery of LOTUS domain proteins EGGD-1 and EGGD-2 by the author's lab and the Gunsalus lab (MIP-1 and MIP-2) described the role of these proteins in promoting the perinuclear localization of P granules and promoting fertility. Here, the authors expand that analysis and further show that the loss of *eggd-1* leads to the mislocalization of other PZM subgranules, and EGGD-1 is essential for the PRG-1/piRNA-mediated silencing and production of specific types of endogenous siRNAs. Depletion of *eggd-1* or other P granule components causes an increase in spermatogenic transcripts and the nuclear accumulation of the conserved transcription factor HLH-30 in the soma, which enhances the transcription of a cuticle-related gene in somatic tissues. The study's results reveal the essential roles of EGGD-1 in small RNA biogenesis and gene regulation and highlight the communication between germ granules and soma.

This study follows a systematic and thorough analysis of EGGD-1 phenotypes, but some of these observations were touched on in the previous EGGD-1/MIP-1 studies, reducing the impact of these findings. Mechanisms beyond P granule assembly and small RNA expression are left unresolved. The findings' relevance is unclear outside the *C. elegans* system and may be more appropriate for a specialized journal.

Other critiques are as follows:

- 1) PGL-1 dispersal by EGGD-1/MIP-1 was already reported in the Cipriani 2021 study.
- 2) The Cipriani study also showed that germ granule components (specifically GLH-1) increase in the cytoplasm of germ cells, much like the increase of PGL-1 in the rachis reported here.
- 3) RFP-tagged PGL-1 constructs have a strong propensity for aggregation, even in wild-type backgrounds, while GFP-tagged PGL-1 does not. For example, a micropublication from Ubel & Phillips reported on the effect of different fluorescent reporters on PGL granules, which may necessitate fluor-switching to determine if the accumulation of PGL in the rachis (or ZNFX-1 at the periphery of these large granules) in *eggd-1* mutants is not an artifact of the RFP tag.

Reviewer #3 (Remarks to the Author):

In this manuscript, Price et al delve further into the phenotypes associated with loss of *eggd-1*. In previous work, this group has shown that *eggd-1* and *eggd-2* are critical for germ granule formation at the nuclear periphery. Here they further the germ granule subcompartments (Z granule, mutator foci) and address the small RNA and mRNA expression changes resulting from *eggd-1* germ granule disruption. For small RNAs, they observe a substantial loss of WAGO/HRDE-1/Mutator/PRG-1-dependent small RNAs and an increase in ALG-3/4 small RNAs and male piRNAs. Consistently, *eggd-1* mutants are also defective in piRNA-mediated silencing. CSR-1 22G-RNAs are unchanged. The increase in male-specific small RNAs also correlates to an increase in spermatogenic transcripts. This increase in spermatogenic transcripts is similar to what has been seen in other small RNA pathway mutants, but it is still unclear whether this is a failure to turn off these genes at the L4-adult transition or whether they are turned on in the adult. Lastly, they show that disruption of germ granules by

eddg-1 leads to increased nuclear HLH-30 expression in somatic cells, leading to upregulation of at least one collagen gene. Overall this paper is clear and well-written and contributes to understanding how germ granule disruption affects gene expression in both somatic and germ cells.

Minor comments -

In Fig 1A, some germ granules are still associated with the nuclear periphery, including some enlarged PGL-1 aggregates. Are these aggregates still associated with nuclear pores?

Fig 2A – legend indicates that diagonal grey lines are 2-fold up and down regulation, but they line up with the red and green dots that are 4-fold up and down regulated. Are the lines actually at 4 fold?

Fig 2E – legend typo “black like” should be “black line”

Since not all spermatogenic RNAs are upregulated – any clues as to differences between the upregulated genes and those that remain unchanged?

Similar to Fig 5C, can you make a Venn diagram of upregulated genes and HLH-30 target genes? Are all of the collagen genes shown in S5B also HLH-30 targets?

On pg 15, second paragraph, “Three silencing pathways engaging cytoplasmic, perinuclear, and nuclear WAGOs, are thought to promote robust exogenous RNAi responses” – be more specific about what pathways you are referring to.

Is there any evidence of germ cells differentiating into sperm instead of oocytes in eddg-1 mutant that could explain increased expression of spermatogenic genes?

REVIEWER COMMENTS

First and foremost, we would like to thank all three reviewers for their constructive and critical comments. These comments have not only aided us in improving our manuscript, but also provide guidance on our future research. In response to reviewers' suggestions, we have conducted additional experiments and revised our manuscript. To facilitate the reviewers' assessment, we have highlighted all the changes using the color blue.

Reviewer #1 (Remarks to the Author):

Germ granules are conserved features of early germ cells that have been postulated to promote germline identity and prevent germ cells from adopting somatic fates. In this work, the authors further examine intrinsically disordered LOTUS domain proteins, EGGD-1, and EGGD-2 (also known as MIP-1 and MIP-2). These proteins have been previously shown by the authors and others to function partially redundantly in perinuclear granule formation in the *C. elegans* germline. That work defined a role for the IDR domains in organizing and anchoring granules to the nucleus and the LOTUS domain in localizing P-granules proximal to the nucleus; however, the other granule types present in germ cells were not examined in detail. Because prior work indicated EGGD-1 supports granule formation but is more important for perinuclear P-granule localization, this study focuses primarily on EGGD-1 functions. Specifically, the authors investigated the importance of proper subcellular localization of various germ cell granules, P-granules, Z-granules, and M- (mutator) granules and how localization contributes to germ cell development.

Prior work suggested that EGGD-1 and 2 act as scaffolds to recruit and regulated developmental transitions by regulating RNA processing machinery and interacting with nuclear pore components. Here loss of *eggd-1* is examined more closely and leads to coalescence of germ granules in embryos and abnormal localization of PZM granules in adults, supporting a role for EGGD-1 in promoting association of various germ granules with nuclear pores or nuclear membrane/tethers granules maintaining granule spatial distribution. Further, they examined the effects on RNA regulation machinery and provide evidence that EGGD-1 is not broadly required for piRNA biogenesis/pathways but is required for PRG-1 silencing and select endogenous siRNAs. In *C. elegans* hermaphrodites, the developing gonad produces sperm first and later switches to oocyte production. Here the authors analysis of transcriptomes and small RNAs provides evidence that RNAs involved in spermatogenesis persist in hermaphrodites in the absence of EGGD-1, suggesting proper granule localization facilitates RNA regulation at this developmental transition. Further, they observed prolonged transgenerational inheritance of RNAi, as also occurs in *prg-1* mutants, one of the improperly localized factors in *egg1* mutants.

By knocking down or mutating different genes required for proper P-granule size or formation, the authors discovered a generalized role of P-granules, rather than a specific role of EGGD-1, in preventing nuclear accumulation of the transcription factor HLH-30 in the adjacent somatic cells and activation of cuticle genes in adjacent somatic

tissues. This surprising observation suggests improper subcellular localization of granules or molecules regulated in these subcellular compartments impacts communication between germline and somatic cells and influences somatic cell development. Overall, the manuscript is well written, and the data are of high quality and are clearly presented. However, a few points related to the exciting observation regarding granules and communication with the soma should be addressed.

Major:

1) A potential role for granules in regulating gene expression in the soma is exciting and unexpected. While investigating the precise mechanism would be beyond the scope of this work, an addressable question that was unclear is whether the soma is detecting a molecule regulated by the perinuclear granules or instead is recognizing that the hermaphrodite germline/oocytes are abnormal or dying. Is the same effect on somatic/cuticle gene expression observed if oocytes are abnormal for reasons unrelated to germ granules or are ablated, or if one ectopically expressed spermatogenesis genes in oocytes?

We'd like to thank the reviewer for this excellent suggestion. We conducted additional experiments to investigate the cause of col-12 upregulation. Our results support the idea that aberrant expression of spermatogenic genes may contribute to the elevated somatic expression of COL-12. More detailed information about these experiments can be found below.

“We next sought to determine the underlying cause of HLH-30 activation and somatic col-12 overexpression in eggd-1 mutants. Previous and current studies have demonstrated the crucial role of EGGD-1 in germline maintenance, oocyte development, and suppression of spermatogenic genes in hermaphrodites (Fig. 5)^{24,26}. This led us to speculate whether defects in oogenesis and/or activation of spermatogenic genes could be responsible for the abnormal col-12 expression. To explore this idea, we generated the sun-1p::rde-1; col-12p::DsRed; rde-1 strain in which the endogenous rde-1 gene is mutated and a single-copy rde-1 transgene is expressed under the germline-specific sun-1 promoter⁹⁰. This strain, which is somatic RNAi deficient but germline RNAi competent, allowed us to examine the effects of germline-specific knockdown⁹⁰. Consistent with a previous report⁹¹, depletion of LIN-41, a TRIM-NHL protein, resulted in a decrease in the size of germ line and complete loss of oocytes (Fig. 7a). However, P granule formation, indicated by PGL-1::RFP expression, appeared unaffected by lin-41 depletion (Fig. 7a). Depletion of lin-41 in sun-1p::rde-1; col-12p::DsRed; rde-1 strain did not alter col-12p::DsRed expression (Fig. 7b), indicating defective oogenesis is not responsible for the col-12 activation. We next induced the expression of male genes in hermaphrodites by depleting key regulatory proteins in the sex determination pathway, including DEAH-box proteins MOG-4 and MOG-5⁷⁹. A previous study has shown that loss of mog-4, or mog-5 leads to the germline masculinization⁷⁹. Indeed, we observed disorganized and masculinized germ line upon the depletion of mog-4 or mog-5 (Fig. 7a). Furthermore, mog-4 knockdown significantly reduced PGL-1::RFP levels, while mog-5 knockdown modestly reduced PGL-1::RFP expression (Fig. 7a). Upon exposure to mog-4 or mog-5 dsRNAs, the sun-1p::rde-1; col-12p::DsRed; rde-1 reporter displayed increased DsRed fluorescent signals (Fig. 7b). Taken together, our findings suggest that aberrant

expression of spermatogenic genes in hermaphrodites may contribute to the elevated somatic expression of col-12.” (Page 15-16).

We also generated *sun-1p::rde-1; eggd-1; col-12p::DsRed; rde-1* quadruple mutants. Depletion of *fem-1* or *fem-3* in this strain did not suppress *col-12p::DsRed* expression. However, there are at least two caveats: 1) the RNAi treatment made worms very sick; 2) we cannot be certain that *fem-1* or *fem-3* knockdown fully suppressed spermatogenic genes in the reporter strain. In an effort to address these concerns, we also used temperature-sensitive *fem-3* (e2006) allele, but found that shifting the *col-12p::DsRed* reporter to 25 °C elevated *col-12p::DsRed* expression even in the wild-type background. We decided to leave these results out, as they did not provide any novel insights.

2) The methods corresponding to the reporter analyses, including argonauts, HLH-30, Col-12p are not clear or are missing from the methods section. How were these signals normalized across conditions?

Thanks for pointing this out. We have revised the main text and updated the methods.

“Given that AGO proteins exhibit varying expression levels, we adjusted the exposure times and laser intensities accordingly to capture the individual GFP::AGO proteins, while maintaining the identical experimental conditions between the wild-type and eggd-1 strains.” (Page 9).

“Quantification of Argonaute protein expression

GFP::PRG-1, GFP::WAGO-1, GFP::WAGO-9/HRDE-1, and GFP::CSR-1 samples were prepared following the procedures described in the Microscopy section. Z stacks were acquired using the 60x water objective with 1 μm step size. Different exposure times and laser intensities were applied to capture individual GFP::AGOs, while the identical parameters were applied between the wild-type and paired eggd-1 strains. In ImageJ, the background signal was determined by selecting an area outside of animals, and subsequently subtracted from the images. To quantify Argonaute protein levels, germ nuclei on the surface of the gonad were selected and the mean intensity of GFP signals were measured. This process was repeated for 25 nuclei per animal. To access the ratio of rachis signals over to total signals, cross sections were extracted from Z stacks. The entire pachytene and rachis regions were selected and integrated densities were measured to quantify the relative abundance of rachis signals in comparison to the overall signal intensity.” (Page 29).

“Imaging and processing of col-12p::DsRed, hlh-30::gfp and piRNA sensor

The reporters in wild-type and mutant backgrounds, as well as those subjected to with RNAi treatments, were prepared following the procedures outlined in the Microscopy section. Z stacks were acquired using a 60x water objective with a 1 μm step size. Using ImageJ, the background signal was determined by measuring the average intensity of an area outside of the worm, and this background signal was subtracted from all z planes. An average intensity projection was used to capture col-12p::DsRed signals. A single plane from the z stack was used to create images of HLH-30::GFP. A maximum intensity projection was created to capture signals of piRNA sensors.” (Page 29-30).

Minor:

In some cases, the numbers of animals/cells examined is unclear because the data are not consistently presented. In some figures numbers are found on the panels or in the legends and in other figures the numbers are in the methods. It would be helpful to present these numbers more consistently.

All authors agree that this is a good suggestion. We decide to include numbers and statistics in the figures and figure legend when applicable. In addition, p-values are consistently presented by asterisks in all boxplots in the revision. Modified figures are summarized below:

1. Figure 2a, add the number of transcripts with upregulated and downregulated 22G-RNAs.
2. Figure S1b, c, d, e, include number of foci and use asterisks to present p-values.
3. Figure 4c, d, numbers are added in figure legend.
4. Figure 6b-d, numbers are added in figure legend.

The abbreviation PZM is used in the introduction but is only fully introduced later. The reviewer is correct. We updated the text in the revised manuscript.

“In C. elegans, germ granules exhibit a distinctive organization into perinuclear sub-compartments. So far at least four types of sub-compartments have been discovered: P granules, Z granules, SIMR foci, and Mutator foci^{11,15-17}. Although these sub-compartments are in close proximity to one another, genetic analyses and proteomic studies revealed distinct components within each sub-compartment^{3,6}. Specifically, P granules contain RNA binding protein PGL-1, Piwi proteins (known as PRG-1 in C. elegans) and piRNAs (Piwi-interacting RNAs)¹⁸⁻²⁰. Z granules, on the other hand, are characterized by the presence of Argonaute protein WAGO-4 (Worm Argonaute protein) and the RNA helicase ZNFX-1^{9,16,21,22}. SIMR foci exhibit the enrichment of the tudor domain protein SIMR-1¹⁷. Mutator foci harbor several mutator proteins, including MUT-7 and MUT-16, which play essential roles in the generation of endogenous siRNAs (small interfering RNAs)^{15,23}.” (page 3).

“we show that loss of eggd-1 leads to coalescence of germ granules and mis-localization of P granules, Z granules, and Mutator foci, collectively referred to as PZM granules, within the adult gonad.” (page 4).

Reviewer #2 (Remarks to the Author):

The recent discovery of LOTUS domain proteins EGGD-1 and EGGD-2 by the author's lab and the Gunsalus lab (MIP-1 and MIP-2) described the role of these proteins in promoting the perinuclear localization of P granules and promoting fertility. Here, the authors expand that analysis and further show that the loss of eggd-1 leads to the mislocalization of other PZM subgranules, and EGGD-1 is essential for the PRG-1/piRNA-mediated silencing and production of specific types of endogenous siRNAs. Depletion of eggd-1 or other P granule components causes an increase in spermatogenic transcripts and the nuclear accumulation of the conserved transcription

factor HLH-30 in the soma, which enhances the transcription of a cuticle-related gene in somatic tissues. The study's results reveal the essential roles of EGGD-1 in small RNA biogenesis and gene regulation and highlight the communication between germ granules and soma.

This study follows a systematic and thorough analysis of EGGD-1 phenotypes, but some of these observations were touched on in the previous EGGD-1/MIP-1 studies, reducing the impact of these findings. Mechanisms beyond P granule assembly and small RNA expression are left unresolved. The findings' relevance is unclear outside the *C. elegans* system and may be more appropriate for a specialized journal.

Thanks for the comments. The first part of comments is addressed below. Regarding the second part, we believe that *C. elegans* is an excellent model for studying germ granules. Recent research on *C. elegans* has provided significant insights into the organization of germ granule and biomolecular condensates. Furthermore, EGGD-1/MIP-1 belongs to a highly conserved protein family that contains the LOTUS domain and are expressed in the animal germ line. Cipriani et al., and our studies have demonstrated EGGD-1/MIP-1 acts as a scaffold for P granule assembly. However, this study is distinct from previous work, as it emphasizes the impact of germ granules on small RNA expression and transcriptome. We have discussed both points in the revised manuscript.

“LOTUS domain containing proteins have emerged as pivotal regulators in germ granule organization. Oskar, containing both LOTUS and OSK domains, organize germ plasms in Drosophila⁹². TDRD5 and TDRD7 possess LOTUS and Tudor domains and are indispensable for germ granule formation in mice⁹³⁻⁹⁵. Recent studies conducted on C. elegans established an essential role of LOTUS domain proteins in facilitating perinuclear localization of germ granules^{24,26,96}.” (Page 16-17).

“Our study, which combines small RNA profiling and genetic experiments, suggests that the perinuclear localization of germ granules promotes small RNA biogenesis and transcriptome surveillance. A specific set of AGO proteins and key RNA processing enzymes are enriched at C. elegans germ granules^{3,4}.” (Page 17).

Other critiques are as follows:

1) PGL-1 dispersal by EGGD-1/MIP-1 was already reported in the Cipriani 2021 study. Cipriani et al., reported PGL-3 dispersal in *mip-1/eggd-1* mutants, while our previous study reported the dispersal of PGL-1. It is important to clarify that the intention of our current study was not to repeat these experiments. Rather, the examination of PGL-1 served two specific purposes: 1) to utilize it as a benchmark for comparing ZNFX-1 and MUT-16, and 2) to investigate the underlying mechanism behind the formation of large cytoplasmic PGL-1 foci. Neither of these two points were addressed in previous studies.

We have properly cited Cipriani et al., 2021 (Page 4, 5, 9, 11, 16, 17 and 18) and highlighted their findings *“Of note, the dispersal of PGL-3, another member of the PGL family, was previously observed when mip-1/eggd-1 was depleted²⁶”* (Page 5) in the revision.

2) The Cipriani study also showed that germ granule components (specifically GLH-1) increase in the cytoplasm of germ cells, much like the increase of PGL-1 in the rachis reported here.

This point is related to the first critique. We acknowledge the reviewer's critique that the Cipriani et al. has reported the change of P granule components including PGL proteins and GLH-1, so did our previous work. However, the focus of our current study was not to replicate or reiterate those findings. Instead, we aimed to investigate the effects of EGGD-1 on Z granules and Mutator Foci, both of which have distinct components and functions compared to P granules. Here we showed that ZNFX-1 foci, PGL-1 and ZNFX-1 granules remained largely separated in the *eggd-1* rachis, indicating demixing of germ granules into sub-compartments is independent of their association with the nuclear membrane. In addition, our work delved into the mechanism underlying the formation of large PGL-1 aggregates in the rachis. It is our hope that the reviewer would agree these new findings merit reporting.

3) RFP-tagged PGL-1 constructs have a strong propensity for aggregation, even in wild-type backgrounds, while GFP-tagged PGL-1 does not. For example, a micropublication from Ubel & Phillips reported on the effect of different fluorescent reporters on PGL granules, which may necessitate fluor-swapping to determine if the accumulation of PGL in the rachis (or ZNFX-1 at the periphery of these large granules) in *eggd-1* mutants is not an artifact of the RFP tag.

We thank the reviewer for this excellent suggestion. The fluorescent tags can change the dynamics of P granule proteins. We therefore generated and analyzed the fluor-swapped strains. New findings are described in the main text, Figs. 1a, b and Supplementary Fig. 1 b, c, and f.

*“Given that the dynamics of PGL-1 foci can change when tagged with different fluorescent proteins³¹, we inspected both PGL-1::RFP and PGL-1::GFP fluorescence on the surface and at the rachis of adult gonad. In wild-type animals, PGL-1::RFP or PGL-1::GFP foci were primarily associated with the periphery of germ cell nuclei (Fig. 1a, b and Supplementary Fig. 1a). However, when *eggd-1* was deleted in either *pgl-1::gfp* or *pgl-1::rfp* strains, it resulted in the dispersal of perinuclear PGL-1 and accumulation of PGL-1 aggregates at the rachis (Fig. 1a, b)²⁴.”* (page 5)

*“We also noted a similar expression pattern for PGL-1::GFP foci. The mean volume of PGL-1::GFP granules at the *eggd-1* nuclear periphery was 2.77-fold smaller than that of wild-type animals, reducing from 0.332 μm^3 to 0.120 μm^3 (Supplementary Fig. 1c). While large PGL-1::GFP foci were observed at the *eggd-1* rachis, their size was smaller than cytoplasmic PGL-1::RFP foci (Supplementary Fig. 1b, c).”* (page 6)

*“Considering the impact of fluorescent tags on protein phase-transition properties³¹, we proceeded to evaluate the organization of PGL-1::GFP and RFP::ZNFX-1 granules. PGL-1::GFP and RFP::ZNFX-1 remained separate at the nuclear peripheries of wild-type and *eggd-1*, as well as at the *eggd-1* rachis (Supplementary Fig. 1f). These findings suggest*

that P granules and Z granules demix into distinct sub-compartments, regardless of their association with the nuclear membrane.” (Page 6).

Reviewer #3 (Remarks to the Author):

In this manuscript, Price et al delve further into the phenotypes associated with loss of *eggd-1*. In previous work, this group has shown that *eggd-1* and *eggd-2* are critical for germ granule formation at the nuclear periphery. Here they further the germ granule subcompartments (Z granule, mutator foci) and address the small RNA and mRNA expression changes resulting from *eggd-1* germ granule disruption. For small RNAs, they observe a substantial loss of WAGO/HRDE-1/Mutator/PRG-1-dependent small RNAs and an increase in ALG-3/4 small RNAs and male piRNAs. Consistently, *eggd-1* mutants are also defective in piRNA-mediated silencing. CSR-1 22G-RNAs are unchanged. The increase in male-specific small RNAs also correlates to an increase in spermatogenic transcripts. This increase in spermatogenic transcripts is similar to what has been seen in other small RNA pathway mutants, but it is still unclear whether this is a failure to turn off these genes at the L4-adult transition or whether they are turned on in the adult. Lastly, they show that disruption of germ granules by *eggd-1* leads to increased nuclear HLH-30 expression in somatic cells, leading to upregulation of at least one collagen gene. Overall this paper is clear and well-written and contributes to understanding how germ granule disruption affects gene expression in both somatic and germ cells.

Minor comments -

In Fig 1A, some germ granules are still associated with the nuclear periphery, including some enlarged PGL-1 aggregates. Are these aggregates still associated with nuclear pores.

We thank the reviewer for raising this interesting idea. In order to investigate it, we generated strains expressing *NPP-11::wrmScarlet*; *PGL-1::GFP* and examined their expression in both wild-type and *eggd-1* mutants. The updated results and figures are presented below:

“when eggd-1 was deleted in either pgl-1::gfp or pgl-1::rfp strains, it resulted in the dispersal of perinuclear PGL-1 and accumulation of PGL-1 aggregates at the rachis (Fig. 1a, b)²⁴. The remaining PGL-1::GFP foci at the eggd-1 nuclear periphery appeared to associate with the nuclear pore protein NPP-11 (Supplementary Fig. 1a)³².” (Page 5)

Fig 2A – legend indicates that diagonal grey lines are 2-fold up and down regulation, but they line up with the red and green dots that are 4-fold up and down regulated. Are the lines actually at 4 fold?

We are grateful to the reviewer for pointing out this labeling issue. The diagonal grey lines presented log₂FC of 2 and -2 which are equivalent to 4 and ¼ fold change respectively. To improve clarity, we have revised the legend to better communicate this information (Page 21 and 22).

Fig 2E – legend typo “black like” should be “black line”

Thanks for pointing this out. We have corrected the typo (Page 21). We made additional changes in figure legend, all of which were highlighted in blue color.

Since not all spermatogenic RNAs are upregulated – any clues as to differences between the upregulated genes and those that remain unchanged?

This is an intriguing idea. We conducted further data analysis. Without applying fold-change cutoff (such as 4-fold upregulation or downregulation), we observed a strong positive correlation between the fold-change of transcripts in *eggd-1* vs. wild-type and the fold-change of transcripts in male vs. female. New findings are updated and presented below:

*“It was intriguing that certain spermatogenic genes were significantly upregulated while some appeared unchanged in *eggd-1* mutants (Fig. 5a, c). To further investigate this finding, we plotted the fold-change of transcripts in *eggd-1* relative to wild-type against the fold-change of transcripts in male (*fem-3*) compared to female (*fog-2*) (Supplementary Fig. 5a). This analysis revealed a strong positive correlation (Pearson correlation, $r = 0.68$, p -value < 0.05). Transcripts highly enriched in male (*fem-3*) displayed significant upregulation in *eggd-1* compared to wild-type, while transcripts with modest enrichment in male (*fem-3*) showed modest or no change (Supplementary Fig. 5a). These global analyses revealed a general upregulation of spermatogenic genes in *eggd-1* mutants, which we further validated by examining individual genes. For instance, *alg-3*, which encodes a male-specific AGO, and *gsp-3*, which encodes a sperm-specific PP1 phosphatase are exclusively expressed in male (*fem-3*) (Fig. 5d, e and Supplementary Fig. 5a). Both genes showed strong upregulation in *eggd-1* mutants (Fig. 5d, e and Supplementary Fig. 5a)^{55,56,77}. In contrast, *spe-44*, a gene encoding a transcription factor⁷⁸, is highly expressed in male (*fem-3*) but also modestly expressed in female (*fog-2*). We found that the expression of *spe-44* was unaffected upon loss of *eggd-1* (Supplementary Fig. 5a, b).” (Page 12-13).*

Similar to Fig 5C, can you make a Venn diagram of upregulated genes and HLH-30 target genes? Are all of the collagen genes shown in S5B also HLH-30 targets?

Thanks for the suggestion. We conducted further analyses. The findings are updated in the Results section (See below and Page 14) and Method section (“Definition of HLH-30 targets”, Page 34).

*“This unbiased search revealed that transcription factors DAF-16 and HLH-30 occupy the *col-12* promoter under the physiological condition (Supplementary Fig. 6a). By analyzing published HLH-30 CHIP-seq data⁸³, we defined 5,158 potential targets of HLH-30. The promoters of 29 genes that are associated with cuticle development exhibited binding of HLH-30 (Supplementary Fig. 6b). Additionally, in our analysis of transcription factor binding site enrichment⁸⁰, we noted a significant overrepresentation of the HLH-30 binding motif in the dataset of genes upregulated in *eggd-1* mutants ($n=1,024$) (Supplementary Fig. 6c). Among the upregulated gene set, the promoters of 94 genes displayed HLH-30 occupancy (Supplementary Fig. 6b).”*

On pg 15, second paragraph, “Three silencing pathways engaging cytoplasmic, perinuclear, and nuclear WAGOs, are thought to promote robust exogenous RNAi responses” – be more specific about what pathways you are referring to. Information on WAGOs and citations are updated.

“Three silencing pathways engaging nuclear WAGO protein HRDE-1, perinuclear WAGO-1 and WAGO-4, and cytoplasmic WAGO proteins including RDE-1 and PPW-1, are thought to promote robust exogenous RNAi responses^{9,22,34,38,97}” (page 17).

Is there any evidence of germ cells differentiating into sperm instead of oocytes in *eggd-1* mutant that could explain increased expression of spermatogenic genes?

We thank the reviewer for the question. We conducted additional experiments to address it. The details of these experiments are updated the Results section (See below and Page 13) and Method section (“Analysis of masculinization of the germ line phenotype”, Page 30-31).

*“In light of the upregulation of spermatogenic genes, we conducted experiments to determine whether *eggd-1* mutants exhibited the MOG phenotype (masculinization of the germ line). We cultured synchronized wild-type and *eggd-1* L1 larvae and subsequently examined the germ line of L4/adult worms using brightfield microscopy as well as fluorescence microscopy with DAPI staining to aid in the visualization of sperm. As a positive control, we employed RNAi against *mog-4*, a gene encoding a DEAH-Box protein⁷⁹. We used the following criteria to determine the MOG phenotype: 1. Completion of vulval development. 2. Absence of oocytes, and 3. Presence of excess sperm in the germ line. While the majority of *mog-4* dsRNA treated animals displayed the MOG phenotype, none of the wild-type animals exhibited it (Fig. 5f, g). Out of the 570 *eggd-1* worms that were imaged, we only identified one hermaphrodite exhibiting the MOG phenotype and one male (Fig. 5f, g). These findings suggest that the upregulation of spermatogenic genes in *eggd-1* mutants is unlikely to be a result of the masculinization of the germ line.”*

REVIEWERS' COMMENTS

Reviewer #1 (Remarks to the Author):

This is a revised manuscript reporting investigation of the importance of proper subcellular localization of various germ cell granules, P-granules, Z-granules, and M- (mutator) granules and how localization contributes to germ cell development. The authors were responsive to prior concerns and addressed each point thoughtfully with additional experiments and or clarifications to the methods and text. Kudos to the authors who conducted new experiments and analyses that strengthen and clarify conclusions as compared to the prior submission. I have no further concerns. Overall, the manuscript is well written, and the data are of high quality and are clearly presented. The findings extend prior work, and are exciting, particularly regarding granules influencing communication and gene expression within the soma, and will be of interest to cell biologists, reproductive, developmental, and stem cell biologists, and those interested in tissue interactions and intercellular communication.

Reviewer #2 (Remarks to the Author):

The authors have thoroughly addressed all the points I brought up. I especially value the addition of the fluor-swapped PGL-1 experiments and their quantification in figure 1 and sup1. These results strengthen the validity of their findings and the stated conclusions.

Reviewer #3 (Remarks to the Author):

The authors have addressed all of my concerns and the manuscript now appears ready for publication.